# Spectral Attention Steering for Prompt Highlighting

**Weixian Waylon Li**[1]**, Yuchen Niu**[2]**, Yongxin Yang**[4]**, Keshuang Li**[3]**,**
**Tiejun Ma**[1]**, Shay B. Cohen**[1]
[1]University of Edinburgh, UK   [2]RayNeo, China   [3]Huawei Research Ltd., UK
[4]Queen Mary University of London, UK
{waylon.li,tiejun.ma}@ed.ac.uk   scohen@inf.ed.ac.uk

## Abstract

Attention steering is an important technique for controlling model focus, enabling capabilities such as *prompt highlighting*, where the model prioritises user-specified text. However, existing attention steering methods require explicit storage of the full attention matrix, making them incompatible with memory-efficient implementations like FlashAttention. We introduce Spectral Editing Key Amplification (*SEKA*), a training-free steering method that tackles this by directly editing key embeddings before attention computation. *SEKA* uses spectral decomposition to steer key embeddings towards latent directions that amplify attention scores for certain tokens. We extend this to Adaptive SEKA (*AdaSEKA*), a query-adaptive variant that uses a training-free routing mechanism to dynamically combine multiple expert subspaces based on the prompt's semantic intent. Our experiments show both methods significantly outperform strong baselines on standard steering benchmarks while adding much lower latency and memory overhead, in compatibility with optimised attention.

## 1 Introduction

The ability to precisely guide the behaviour of large language models (LLMs) is paramount as they are increasingly deployed in high-stakes domains. This broad field of model steering encompasses various techniques, from activation steering, which aims to control high-level semantic attributes like style or factual recall by intervening in MLP layers (Subramani et al., 2022; Turner et al., 2023; Qiu et al., 2024; Turner et al., 2024; Wang et al., 2025; Stolfo et al., 2025), to attention steering, which operates at a more granular level to direct the model's focus to specific tokens within a prompt. This paper focuses on the latter, where prompt highlighting is one of the key applications. Current state-of-the-art methods, such as PASTA (Zhang et al., 2024), operate by editing the attention score matrix after it has been computed. This post-hoc manipulation creates a critical bottleneck: it requires computing the full attention matrix, making these methods incompatible with modern, IO-aware implementations like FlashAttention (Dao et al., 2022; Dao, 2024) that are essential for efficient processing. This architectural limitation, coupled with the need for costly, task-specific searches to identify which attention heads to steer, makes them less practical.

In this paper, we propose to intervene in the input of the attention mechanism rather than edit its output. We introduce Spectral Editing Key Amplification (*SEKA*), a novel, training-free framework that steers attention by directly modifying key vectors before the attention scores are calculated. Our core insight is that we can learn a universal "relevance subspace" for a given task by applying spectral decomposition to key embeddings derived from contrastive prompts. These learned directions are then used to construct a projection matrix that amplifies the relevant features of highlighted keys via a simple, geometrically interpretable transformation: $\boldsymbol{k}' = \boldsymbol{k} + g\boldsymbol{P}\boldsymbol{k}$.

Additionally, we propose Adaptive SEKA (*AdaSEKA*), an advanced variant that learns a bank of task-specific "expert" projections (e.g., for factual recall versus instruction following). At inference time, *AdaSEKA* uses a computationally cheap, training-free routing mechanism to create a dynamic, query-aware steering operator by blending these experts based on the prompt's semantic intent. Our

method is fully compatible with FlashAttention as it operates directly on the key embeddings with negligible computational overhead.

Our experiments confirm the effectiveness of this approach. Both *SEKA* and *AdaSEKA* achieve superior results on standard benchmarks for knowledge conflicts, occupation extraction, and instruction following. Furthermore, *AdaSEKA*'s query-adaptive routing mechanism demonstrates superior performance by dynamically tailoring the steering to the prompt's semantic intent. Crucially, we show that these performance gains are achieved with negligible overhead. *SEKA* adds only $\approx 0.03$s of latency per sample, in stark contrast to comparable methods like PASTA which incur a +1.03s inference time and nearly double the memory usage.

## 2 PROBLEM DEFINITION AND MOTIVATIONS

In this section, we formalise the problem of *prompt highlighting* as an instance of attention bias and present the motivation for our spectral attention steering approach, which aims to address the limitations of existing methods.

**Problem Definition.** Given a prompt $\boldsymbol{x} = (x_1, \ldots, x_T)$ consisting of $T$ tokens, with a subset of token indices $\mathcal{H} \subset \{1, \ldots, T\}$ identifying the *highlighted* tokens (in practice, surrounded by markers such as $\star\star$), our goal is to steer the attention of the model so that these tokens receive increased focus from queries. In standard multi-head attention, the unnormalised attention score between query $i$ and key $j$ is $\text{Attn}(i, j) = \frac{\boldsymbol{q}_i^\top \boldsymbol{k}_j}{\sqrt{d_k}}$, where $\boldsymbol{q}_i, \boldsymbol{k}_j \in \mathbb{R}^{d_k}$ are the query and key vectors, and $d_k$ is the head dimension.

**Objective.** We aim to amplify the attention assigned to highlighted tokens by introducing an additive, controllable term to the attention score for each $(i, j)$ where $j \in \mathcal{H}$: $A'_{ij} = A_{ij} + \Delta_{ij}$, where $\Delta_{ij}$ is designed to selectively boost the attention towards user-specified highlighted tokens.

**Motivation.** Existing approaches typically modify attention after it has been computed. For example, PASTA (Zhang et al., 2024) rescales rows of the attention matrix as shown in equation 1, where $C_i$ is a row normalisation factor and $\alpha > 1$ scales attention to highlighted tokens.

$$[T(\boldsymbol{A})]_{ij} = \begin{cases} \alpha \dfrac{A_{ij}}{C_i}, & \text{if } j \in \mathcal{H}, \\ \dfrac{A_{ij}}{C_i}, & \text{otherwise.} \end{cases} \tag{1}$$

Similarly, positional calibration methods such as Found-in-the-Middle (Hsieh et al., 2024) subtract a baseline from the positional attention bias. Let $x_k$ denote the position of the $k$-th token, and $\text{Attn}_{\text{ori}}(x_k)$ the original positional bias. The calibrated bias is $\text{Attn}_{\text{calibrated}}(x_k) = \text{Attn}_{\text{ori}}(x_k) - \text{Attn}_{\text{baseline}}(x_k)$, where $\text{Attn}_{\text{baseline}}(x_k)$ is estimated independently of content relevance.

Both strategies require explicit storage of the full attention matrix, which is incompatible with memory-efficient implementations such as FlashAttention (Dao et al., 2022; Dao, 2024). Moreover, methods like PASTA often rely on costly head search to decide which attention heads to steer. These limitations motivate the consideration for an alternative steering mechanism that operates *before* attention scores are computed, avoiding any need to materialise or modify the attention matrix. Since attention depends on query–key inner products, equivalent control can be achieved by editing either representation (shown in Section 3.2). Given our objective of amplifying attention to a specific subset of tokens $\mathcal{H}$, key-side intervention is the natural choice: the key vector $\boldsymbol{k}_j$ is indexed by token position $j$ and therefore governs how strongly each individual token is attended to.

To provide empirical evidence on whether such a pre-attention intervention is feasible, we analyse how key representations change under shifts in contextual relevance. We first construct synthetic contrastive prompt triplets under three conditions: (1) *neutral* (context only), (2) *positive* (context aligned with a relevant query), and (3) *negative* (context paired with an irrelevant query). The construction of such synthetic triplets is described in Appendix A.

Using the Qwen3-1.7B-Base model (28 layers, 8 heads), we extract the key embeddings corresponding to the same token spans under both positive and negative prompts for each (layer, head) pair. We

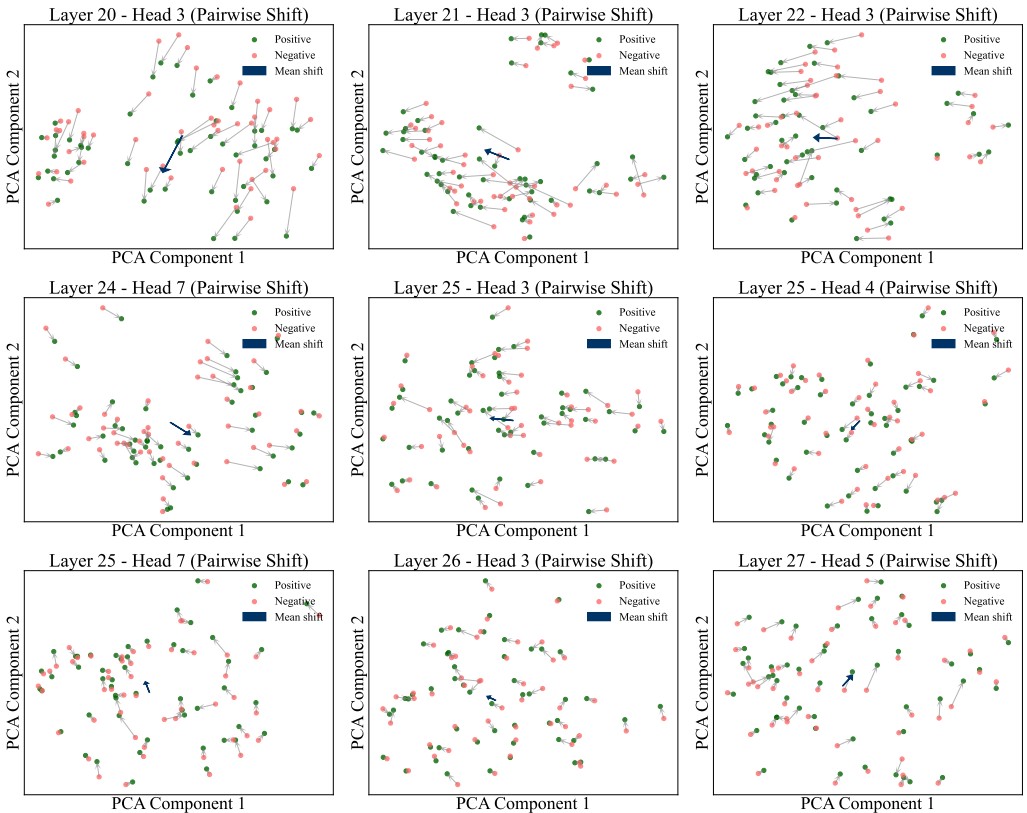

Figure 1: Visualisation of pairwise key embedding shifts across different (layer, head) in Qwen3-1.7B-Base via PCA. Positive vs. negative representations are plotted for 26 shared token spans. Grey arrows trace individual shifts; the dark blue arrow shows the average displacement.

then apply PCA to jointly project these paired embeddings into two dimensions, and visualise the result using a combination of scatter plots and directed arrows. Each arrow originates from a negative key and points to its corresponding positive key, capturing the pairwise representational shift induced by changing question relevance. To summarise the overall trend, we also plot the mean shift vector across all pairs. Figure 1 shows that certain heads exhibit robust and consistent directional shifts in key embeddings when token relevance changes. Each plot visualises 26 key embedding pairs corresponding to shared token spans, extracted from 10 positive–negative prompt pairs.

These findings suggest that relevance is encoded in a structured subspace of key representations, motivating our approach that edits key embeddings before attention is computed: $\boldsymbol{k}'_j = \boldsymbol{k}_j + g\boldsymbol{P}\boldsymbol{k}_j$, where $\boldsymbol{P}$ is a projection matrix (defining a relevance subspace per key-value head), and $g$ is a scaling coefficient. This preserves compatibility with efficient attention implementations while providing a geometrically interpretable mechanism for steering attention towards highlighted tokens.

## 3 SPECTRAL ATTENTION STEERING FOR PROMPT HIGHLIGHTING

As shown in Figure 2, we propose a new method, *Spectral Editing Key Amplification* (*SEKA*), and its query-adaptive variant, *AdaSEKA*. Both methods achieve prompt highlighting by directly editing key embeddings before the attention computation. The core mechanism of *SEKA* is inspired by the Spectral Editing of Activations (SEA) algorithm (Qiu et al., 2024), adapting it from semantic-level activation steering to the token-wise attention steering required for prompt highlighting.

### 3.1 SPECTRAL LEARNING OF RELEVANCE-ALIGNED PROJECTIONS (OFFLINE)

Using the token-level key embeddings obtained from the aforementioned synthetic contrastive prompts (Section 2 and Appendix A), denoted $\boldsymbol{h}$ (neutral), $\boldsymbol{h}^+$ (positive), and $\boldsymbol{h}^-$ (negative), we

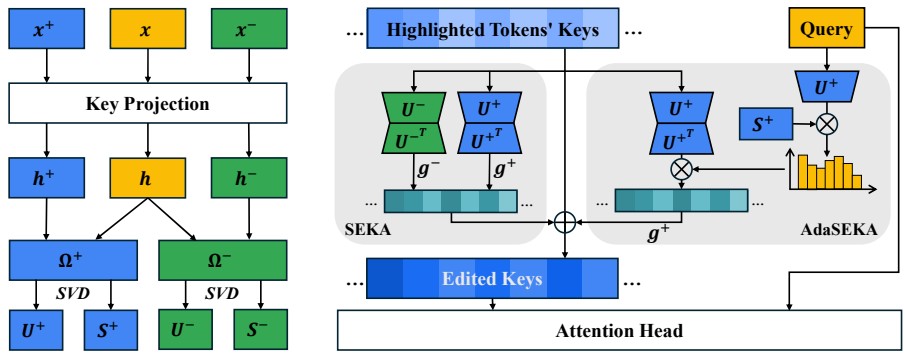

Figure 2: An overview of *SEKA* and *AdaSEKA*. $\boldsymbol{x}$: context; $\boldsymbol{h}$: key embedding; $\boldsymbol{\Omega}$: cross-covariance; $\boldsymbol{U}$: left singular vectors; $\boldsymbol{S}$: singular values; $g$: gain coefficient. *SEKA* applies fixed gains, while *AdaSEKA* uses the query to compute dynamic steering weights.

compute cross-covariance matrices for each transformer layer $\ell$ and key-value head $h$: $\boldsymbol{\Omega}_{\ell,h}^{+} = \frac{\boldsymbol{h}^{\top}\boldsymbol{h}^{+}}{n}$, $\boldsymbol{\Omega}_{\ell,h}^{-} = \frac{\boldsymbol{h}^{\top}\boldsymbol{h}^{-}}{n}$, where $n$ is the number of sampled tokens. Singular value decomposition (SVD) is then applied: $\boldsymbol{\Omega}_{\ell,h}^{+} = \boldsymbol{U}_{\ell,h}^{+}\boldsymbol{S}_{\ell,h}^{+}\boldsymbol{V}_{\ell,h}^{+\top}$, $\boldsymbol{\Omega}_{\ell,h}^{-} = \boldsymbol{U}_{\ell,h}^{-}\boldsymbol{S}_{\ell,h}^{-}\boldsymbol{V}_{\ell,h}^{-\top}$.

In SVD, $\boldsymbol{S}_{\ell,h}^{+}$ and $\boldsymbol{S}_{\ell,h}^{-}$ represent the singular values of the positive and negative cross-covariance matrices, respectively. These singular values quantify the magnitude of cross-covariance captured by each component of the projection. The larger the singular value, the more significant the corresponding singular vector (projection direction) is in explaining the cross-covariance between the token key embeddings.

In equation 2, for the positive projection $\boldsymbol{P}_{\ell,h}^{+}$, we use the *top* singular vectors corresponding to the largest singular values, which capture directions most associated with relevant (highlighted) features. For the negative projection $\boldsymbol{P}_{\ell,h}^{-}$, we use the *least-significant* singular vectors, associated with the smallest singular values, to target directions least associated with relevance.

$$\boldsymbol{P}_{\ell,h}^{+} = \boldsymbol{U}_{\ell,h,:,:k^{+}}^{+}(\boldsymbol{U}_{\ell,h,:,:k^{+}}^{+})^{\top}, \quad \boldsymbol{P}_{\ell,h}^{-} = \boldsymbol{U}_{\ell,h,:,k^{-}:}^{-}(\boldsymbol{U}_{\ell,h,:,k^{-}:}^{-})^{\top}, \tag{2}$$

where $k^{+}$ and $k^{-}$ are chosen such that they capture at least a proportion $\gamma$ of the total singular value sum:

$$\frac{\sum_{i=1}^{k^{+}}\boldsymbol{S}_{\ell,h,i}^{+}}{\sum_{i=1}^{d_{k}}\boldsymbol{S}_{\ell,h,i}^{+}} \geq \gamma, \quad \frac{\sum_{i=1}^{k^{-}}\boldsymbol{S}_{\ell,h,i}^{-}}{\sum_{i=1}^{d_{k}}\boldsymbol{S}_{\ell,h,i}^{-}} \geq \gamma. \tag{3}$$

The threshold $\gamma$ is a hyperparameter that controls how much of the variance in the data we wish to retain when creating the projection matrices. By selecting the top $k^{+}$ singular vectors for the positive covariance and $k^{-}$ for the negative covariance, we capture the most relevant directions in the key embeddings for each type of projection. The learned projectors $\{\boldsymbol{P}_{\ell,h}^{+}, \boldsymbol{P}_{\ell,h}^{-}\}$ are stored per layer and head, enabling fine-grained steering at inference time.

## 3.2 SPECTRAL EDITING FOR HIGHLIGHTED TOKENS (INFERENCE)

During inference, *SEKA* injects the learned projections into key embeddings before attention scores are computed. For clarity, we omit the explicit $(\ell, h)$ indices on key vectors $\boldsymbol{k}_{j}$ and queries $\boldsymbol{q}_{i}$, although they are in practice layer- and head-specific. For each token key $\boldsymbol{k}_{j} \in \mathbb{R}^{d_{k}}$ at layer $\ell$ and head $h$, the edited embedding is defined as:

$$\boldsymbol{k}_{j}' = \boldsymbol{k}_{j} + \frac{g^{+} \cdot \boldsymbol{P}_{\ell,h}^{+}\boldsymbol{k}_{j} + g^{-} \cdot \boldsymbol{P}_{\ell,h}^{-}\boldsymbol{k}_{j}}{2}, \tag{4}$$

where $\boldsymbol{P}_{\ell,h}^{+}, \boldsymbol{P}_{\ell,h}^{-} \in \mathbb{R}^{d_{k} \times d_{k}}$ are the selected projection matrices and $g^{+}, g^{-}$ are two independently adjustable scalars controlling the positive and negative steering gains. All vectors (e.g., $\boldsymbol{k}_{j}$, $\boldsymbol{q}_{i}$,

$\boldsymbol{x}$) are column vectors unless otherwise specified. This adjustment modifies the attention logits as equation 5, where $\boldsymbol{q}_i \in \mathbb{R}^{d_k}$ is the $i$-th query vector. It is algebraically equivalent to augmenting the original attention score matrix $\boldsymbol{A}$ with a low-rank relevance bias matrix $\boldsymbol{B}$:

$$\text{Logits}_{ij} = \frac{\boldsymbol{q}_i^\top \boldsymbol{k}_j}{\sqrt{d_k}} + \frac{\boldsymbol{q}_i^\top \left( \dfrac{g^+ \cdot \boldsymbol{P}_{\ell,h}^+ \boldsymbol{k}_j + g^- \cdot \boldsymbol{P}_{\ell,h}^- \boldsymbol{k}_j}{2} \right)}{\sqrt{d_k}} = \boldsymbol{A}_{ij} + \boldsymbol{B}_{ij}. \tag{5}$$

Thus, *SEKA* can be interpreted as adding a key-dependent term to the attention scores, amplifying each token's the directions aligned with the relevance subspace (detailed in Appendix C). Unlike methods that directly manipulate the attention matrix, *SEKA* achieves equivalent modulation by editing the key vectors themselves, offering a more structured and interpretable mechanism. Moreover, because *SEKA* operates entirely on key representations prior to attention computation, it requires no access to or storage of the attention matrix, making it inherently compatible with memory-efficient implementations like FlashAttention.

## 3.3 Variant: Query-Driven Adaptive SEKA

While the standard *SEKA* framework provides effective token-level attention steering, practical deployment often requires hyperparameter tuning across different tasks and model families due to the static projections. To address this limitation and reduce the need for manual configuration, we introduce *Adaptive SEKA (*AdaSEKA*)*, which automatically selects and combines expert projections based on query-specific relevance signals.

**Multi-Expert Projection Learning.** We extend the projection learning framework to accommodate multiple domain-specific experts. For each expert[1] $m \in \{1, \ldots, M\}$, we constructed samples from datasets $\mathcal{D}_m$ for different tasks. Each expert learns its own set of positive SVD components $\{\boldsymbol{U}_{m,\ell,h}^+, \boldsymbol{S}_{m,\ell,h}^+, \boldsymbol{V}_{m,\ell,h}^+\}$ following the standard *SEKA* procedure. This process results in a set of SVD components for each expert, layer, and head, which can be represented as a 5D tensor ($\boldsymbol{U}^+ \in \mathbb{R}^{M \times L \times H \times d_k \times d_k}$), where $L$ is the number of layers, and $H$ is the number of heads.

**Query-Adaptive Expert Routing.** At inference time, we extract the query vector $\boldsymbol{q}_{\ell,h}$ at layer $\ell$ and head $h$ of the last token in the prompt, as the last token serves as the global aggregator of prompt information and hugely influences the downstream generation (Barbero et al., 2024; Qiu et al., 2024). We then compute dynamic coefficients that determine the contribution of each expert:

$$\alpha_{m,\ell,h}(\boldsymbol{q}_{\ell,h}) = \frac{\sum_{k=1}^K (\boldsymbol{q}_{\ell,h}^\top \boldsymbol{u}_{m,\ell,h}^{+(k)}) \cdot \sigma_{m,\ell,h}^{+(k)}}{\max_m \left| \sum_{k=1}^K (\boldsymbol{q}_{\ell,h}^\top \boldsymbol{u}_{m,\ell,h}^{+(k)}) \cdot \sigma_{m,\ell,h}^{+(k)} \right|}, \tag{6}$$

where $\sigma_{m,\ell,h}^{+(k)}$ is the corresponding $k$-th singular value, and $K$ is the number of top singular components used (typically $K = 5$).

This formulation measures how well the query aligns with each expert's main projection directions, weighted by their singular values. The denominator normalises by the largest absolute alignment across experts, which keeps the coefficients on a comparable scale and preserves whether the alignment is positive or negative.

The final projection matrix at layer $\ell$ and head $h$ is constructed as a weighted combination of expert projections: $\boldsymbol{P}_{\text{dynamic},\ell,h}(\boldsymbol{q}_{\ell,h}) = \sum_{m=1}^M \alpha_{m,\ell,h}(\boldsymbol{q}_{\ell,h}) \cdot \boldsymbol{U}_{m,\ell,h,:,:K}^+ (\boldsymbol{U}_{m,\ell,h,:,:K}^+)^\top$, where $\boldsymbol{U}_{m,\ell,h,:,:K}^+$ denotes the first $K$ columns of $\boldsymbol{U}_{m,\ell,h}^+$, corresponding to the most significant singular vectors.

This approach reconstructs projection matrices on-demand using only the top-$K$ components, providing computational efficiency whilst enabling automatic expert selection. The key transformation during inference becomes: $\mathbf{k}_j' = \mathbf{k}_j + g \cdot \boldsymbol{P}_{\text{dynamic},\ell,h}(\boldsymbol{q}_{\ell,h}) \mathbf{k}_j$.

Crucially, *AdaSEKA* offers several practical advantages: (1) **Reduced configuration effort**: Automatic expert routing reduces the number of hyper-parameters tuning for different tasks and models

---

[1]Experts can vary across task-specific datasets, such as factual correction and instruction-following.

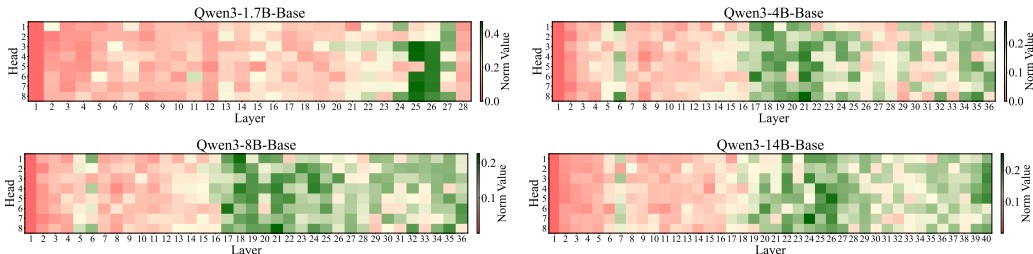

Figure 3: Heatmaps of the average per-token $\ell_2$ distance between positive and negative key embeddings across all KV heads and layers for four Qwen3 model sizes. Higher values (green) indicate greater separation between positive and negative key representations.

(shown in Appendix F). (2) **Modular deployment**: New experts can be integrated without recalculating existing ones. (3) **Interpretable routing**: Expert selection is based on explicit query-expert alignment scores. We derive four expert projections from four distinct datasets. The process of constructing data samples for learning these projections is detailed in Appendix B.

## 3.4 SELECTING RELEVANCE-SENSITIVE KEY-VALUE HEADS

*SEKA* are most effective when applied selectively to KV heads that are naturally sensitive to prompt relevance. As demonstrated in the qualitative visualisations in Figure 1 and discussed in Section 2, the key embedding for a given token span consistently shift in vector space when the question in the prompt is changed from an irrelevant one to a relevant one. In this section, we formalise a method to quantify this relevance sensitivity across all layers and heads to inform our selection strategy.

Figure 3 shows the $\ell_2$ distance between positive and negative key embeddings, averaged over all answer tokens from our synthetic dataset (as defined in Appendix A). This variation is examined across different layers and heads of the Qwen3 model in various sizes.

We observe that the distinction between relevant and irrelevant prompts is not uniform: larger norm values (green) consistently emerge in the mid-to-late layers, while early layers and a subset of heads display minimal shift (red), suggesting the retrieval behaviour is less likely to happen at those layers. This finding is strongly aligned with recent mechanistic analyses. Michel et al. (2019); Voita et al. (2019); Clark et al. (2019); Neo et al. (2024); Li et al. (2023b) highlight that attention modules display various token-attending patterns across different heads. Qiu et al. (2025) demonstrate that retrieval effectiveness relies on only a subset of attention heads, identified via probing and relevance filtering. Wu et al. (2025) further show that this sparse set of "retrieval heads" are almost exclusively located in the mid-to-late layers of the transformer. These heads are intrinsic to the base models, remain consistent after fine-tuning, and are dynamically activated according to the context. Therefore, motivated by this alignment, we restrict projection to only those (layer, head) pairs where the empirical $\ell_2$ difference between positive and negative key embeddings exceeds a threshold. This selective approach ensures that attention steering is concentrated on components empirically associated with retrieval behaviour, while leaving other heads unaffected. In this way, we amplify relevance signals only where necessary, minimising unintended influence on unrelated model components.

Formally, for each layer $\ell$ and head $h$, let $S$ denote the set of all answer tokens (across all samples in the data), with $|S| = N$. The average per-token $\ell_2$ distance is computed as $D_{\ell,h} = \frac{1}{N} \sum_{i=1}^{N} \left\| \boldsymbol{h}_{\ell,h,i}^{+} - \boldsymbol{h}_{\ell,h,i}^{-} \right\|_2$, where $\boldsymbol{h}_{\ell,h,i}^{+}$ and $\boldsymbol{h}_{\ell,h,i}^{-}$ are the positive and negative key embeddings for token $i$ in $S$. Projection is applied only if $D_{\ell,h} \geq \delta_{\min}$, where $\delta_{\min}$ is a tunable hyperparameter tuned via grid search on a validation set (typically in $[0, 0.6]$).

## 4 EXPERIMENTAL SETUP

We consider *SEKA* particularly useful in scenarios that require emphasis or highlighting within the prompt. This includes the tasks used to evaluate PASTA (Zhang et al., 2024), which involve (i) handling complex user instructions (e.g., pronoun rewriting), (ii) interpreting lengthy and noisy contexts (e.g., Bias in Bios; De-Arteaga et al. 2019), and (iii) resolving in-context knowledge conflicts (e.g.,

CounterFact; Meng et al. 2022). In addition, *SEKA* enables us to invert the typical U-shaped performance observed in the "lost in the middle" setting (Liu et al., 2024) by simply highlighting the middle of long contexts, thus improving model recall for these challenging positions.

## 4.1 STANDARD BENCHMARKS FOR ATTENTION STEERING

We follow the standard benchmarks used by PASTA, ensuring consistent selection of highlighted tokens. Table 1 summarises the tasks, prompt formats, and evaluation metrics. The CounterFact task is based on the COUNTERFACT dataset (Meng et al., 2022), while the remaining two tasks (Bias in Bios, Pronouns changing) are derived from the BIASBIOS dataset (De-Arteaga et al., 2019), in line with previous research (Zhang et al., 2024). We enhance the evaluation metric for the Pronouns changing task to address flaws in the original protocol which can misleadingly reward empty response, with the other metrics remaining consistent. Further details, including an introduction to each benchmark task and the calculation of metrics, are available in Appendix E.

Table 1: Summary of standard benchmarks for attention steering. **Tokens in bold** indicate where attention steering is applied.

| Task | Prompts | Metrics |
|------|---------|---------|
| Counterfact | Previously, *[old fact]*. Currently, ***[new fact]***. *[question]*. | Efficacy score (ES), Paraphrase score (PS) |
| Bias in Bios | ***[person's occupation]***. *[career history, may not directly related to prediction]*. *[person]* has the occupation of a/an ___ | Accuracy (Acc.) |
| Pronouns changing | *[biographical contexts]*. **Substitute 'she' and 'he' with 'they' and generate the occupation of** *[person]* **after changing pronouns.** | Pronoun-weighted Lexical overlap Score (P. Score), All-changed P. Score |

**Benchmark Methods.** We begin by using direct prompting of the original model as a baseline. Additionally, we include another baseline that incorporates ** marks around the highlighted context. For attention steering methods, ** is solely used to determine the token indices for steering and is removed from the input IDs. We then benchmark our proposed methods, *SEKA* and *AdaSEKA*, against the existing attention steering method *PASTA*. We also compare with *Selective Prompt Anchoring (SPA)* (Tian & Zhang, 2025), a prompt highlighting method that operates on the logit distributions of the LLMs. Additionally, we evaluate *SEKA* with random projections applied and without the KV heads selector to serve as an ablation study.

## 4.2 U-SHAPE INVERSION IN THE LOST-IN-THE-MIDDLE SETTING

To further examine *SEKA* 's ability to steer model attention to specific regions within a long context, we introduce an additional experiment targeting positional recall in the challenging lost-in-the-middle setting (Liu et al., 2024). This setting refers to the widely observed phenomenon where LLMs exhibit strong recall for information presented at the beginning and end of long contexts, but their performance substantially degrades when the relevant information is located in the middle, resulting in a characteristic U-shaped performance curve. Each of our inputs consists of a long context comprising 30 passages, where only one gold passage contains the true answer to a given question and the rest serve as distractors. The position of the gold passage is varied to test the model's positional sensitivity. Each input is formatted as: "*Context:* \n *[P1 Title]* \n *[P1 Text]* ... *[P30 Title]* \n *[P30 Text]* \n\n *Question: ex['question']* \n *Answer:*".

Unlike prior work that aims to mitigate this effect, our objective is to directly investigate whether explicit relevance highlighting via *SEKA* can invert this U-shaped curve. By steering attention towards the middle passages, we test if the typical performance trough for mid-context answers can be transformed into a peak, providing insight into the controllability of positional recall in LLMs.

**Metrics.** We use exact match (EM) score as the evaluation metric, following Liu et al. (2024): a prediction is considered correct if it contains the ground-truth short answer span. To discourage verbose or off-topic completions, the generated answer is limited to a maximum of 60 tokens.

Table 2: Performance on standard benchmarks. **Bold** = best. Underline = second best. We include two ablation studies for *SEKA*: *"w/o learn"* uses random projections instead of spectrally learned ones, and *"w/o learn&filt"* further removes the head filtering mechanism.

| Model | Metric | Baselines | | | | Our Methods | | | |
|---|---|---|---|---|---|---|---|---|---|
| | | Original | **-marked | PASTA | SPA | *SEKA* | *w/o learn* | *w/o learn&filt* | *AdaSEKA* |
| Qwen3-4B | CounterFact (ES) | 45.00 | 57.70 | 97.16 | 65.24 | **99.02** | 94.96 | 86.12 | 98.90 |
| | CounterFact (PS) | 45.64 | 52.12 | 96.03 | 57.71 | 98.61 | 92.38 | 86.20 | **98.72** |
| | Bias in Bios (Acc.) | 79.84 | 82.94 | 89.58 | 68.00 | 91.02 | 86.62 | 71.76 | **91.86** |
| | Pronoun (P. Score) | 93.14 | 95.76 | **95.82** | 80.27 | 95.18 | 90.42 | 41.98 | 94.54 |
| | Pronoun (A. P. Score) | 90.52 | 93.88 | **94.64** | 78.19 | 93.26 | 88.66 | 36.95 | 92.08 |
| Qwen3-8B | CounterFact (ES) | 39.04 | 56.24 | 92.70 | 69.26 | **99.08** | 96.12 | 95.18 | 99.00 |
| | CounterFact (PS) | 39.59 | 49.80 | 91.68 | 58.76 | 98.96 | 94.74 | 89.69 | **98.97** |
| | Bias in Bios (Acc.) | 76.08 | 80.60 | 86.32 | 37.02 | **88.74** | 87.26 | 74.90 | 88.50 |
| | Pronoun (P. Score) | 98.00 | 98.10 | 98.86 | 72.61 | 98.56 | 98.12 | 80.53 | **99.68** |
| | Pronoun (A. P. Score) | 97.84 | 97.84 | 98.72 | 74.95 | 98.26 | 97.90 | 80.85 | **99.52** |
| Qwen3-14B | CounterFact (ES) | 37.56 | 45.52 | 76.84 | 84.22 | 98.92 | 86.28 | 95.26 | **99.00** |
| | CounterFact (PS) | 36.12 | 40.12 | 66.33 | 76.11 | 99.02 | 88.07 | 92.02 | **99.15** |
| | Bias in Bios (Acc.) | 85.22 | 90.94 | 88.46 | 57.86 | 90.28 | 88.02 | 88.44 | **91.22** |
| | Pronoun (P. Score) | 98.42 | 98.86 | 90.98 | 91.60 | 98.66 | 96.32 | 88.60 | **99.88** |
| | Pronoun (A. P. Score) | 98.22 | 98.68 | 90.94 | 92.20 | 98.54 | 96.36 | 89.76 | **99.86** |
| Gemma3-4B | CounterFact (ES) | 55.04 | 57.56 | 78.36 | 93.90 | 98.04 | 95.14 | 94.46 | **98.74** |
| | CounterFact (PS) | 47.77 | 45.82 | 59.53 | 91.92 | 98.83 | 92.25 | 91.98 | **99.05** |
| | Bias in Bios (Acc.) | 89.90 | 91.00 | 82.58 | 48.02 | 92.42 | 85.60 | 77.16 | **92.92** |
| | Pronoun (P. Score) | 41.34 | 38.86 | 67.39 | 76.05 | 81.53 | 53.58 | 51.78 | **93.76** |
| | Pronoun (A. P. Score) | 35.25 | 32.45 | 66.43 | 74.45 | 81.11 | 48.82 | 51.94 | **93.58** |
| Gemma3-12B | CounterFact (ES) | 45.34 | 48.72 | 68.30 | 93.76 | **98.86** | 63.08 | 60.96 | 92.48 |
| | CounterFact (PS) | 37.21 | 36.69 | 71.72 | 91.24 | **99.27** | 50.59 | 76.37 | 93.65 |
| | Bias in Bios (Acc.) | 91.26 | 92.90 | **94.72** | 46.88 | 93.04 | 91.84 | 90.54 | 91.14 |
| | Pronoun (P. Score) | 93.92 | 95.78 | 68.47 | 86.41 | **97.70** | 47.26 | 55.56 | 96.88 |
| | Pronoun (A. P. Score) | 94.96 | 96.42 | 68.01 | 84.99 | **97.24** | 51.24 | 58.76 | 95.84 |

**Benchmark Methods.** We compare *SEKA* against a standard baseline: directly prompting the base LLM without any intervention, and also *PASTA*. On top of this, we apply *SEKA* in two configurations: (i) steering only the middle region of the context (specifically passages 4 through 25), and (ii) steering all context passages. Although Hsieh et al. (2024) presents another potential baseline, we exclude it due to the unavailability of its code implementation.

## 5 RESULTS

### 5.1 STANDARD BENCHMARKS: *SEKA* PROVIDES EFFICIENT ATTENTION STEERING

The main experimental results are presented in Table 2. We tested the Qwen3 model (Yang et al., 2025) in various sizes, including 4B, 8B, and 14B, as well as the Gemma3 model (Gemma Team, 2025) in sizes of 4B and 12B. For PASTA, we present its best performance from three configurations to ensure a robust comparison (see Appendix H for full details). Furthermore, specific examples and the corresponding outputs from both the original model and *SEKA* are available in Appendix I.

The results demonstrate that *SEKA* and *AdaSEKA*, are highly effective at steering LLM attention, generally outperforming both baseline models (ranked among the top two most of the time) and existing methods across various tasks and model scales. As demonstrated in Section 6, these improvements are achieved with significantly lower overhead compared to PASTA and SPA.

A primary finding is the efficacy of attention-level interventions on tasks requiring factual recall. On CounterFact, both *SEKA* and PASTA achieve near-perfect scores (e.g., 99.02 and 97.16 respectively for Qwen3-4B), validating the general approach of steering attention for knowledge conflicts, while the logit-based SPA lags considerably. Within this effective category, our methods consistently hold a performance advantage. This trend continues in the Bias in Bios task, where *SEKA* and *AdaSEKA* generally secure the top two positions across all models.

Performance on the instruction-following Pronoun Changing task is strongly correlated with the base model's pretrained sensitivity to simple emphasis markers. For the Qwen3 family, which is partially responsive to simple markdown emphasis, the "**-marked" baseline is notably strong. This contrasts with earlier conclusions that LLMs are inherently restricted to processing plain text without

stylistic cues or emphasis markers (Brown et al., 2020; Wei et al., 2022). However, *AdaSEKA* still provides further improvement, delivering SOTA performance (e.g., an A. P. Score of 99.52 on Qwen3-8B). The advantage of our methods is most pronounced on the Gemma3-4B which is less responsive to the markdown emphasis. This demonstrates our method's significant value, especially for smaller models that are less receptive to basic emphasis grammar.

Finally, our ablation studies validate the method's core components. Using random projections with head filtering (*w/o learn*) proves beneficial but is clearly suboptimal, underscoring the value of our spectral learning approach. Removing both the learned projections and the head-filtering mechanism (*w/o learn&filt*) causes a catastrophic decline in performance. For instance, on the Qwen3-4B Pronoun task, the A. P. Score drops from the original 90.52 to 36.95. This conclusively demonstrates that both learning meaningful relevance subspaces and selectively applying them to the appropriate KV heads are essential for success.

## 5.2 LOST IN THE MIDDLE

With the setting described in Section 4.2, we highlight two key findings when benchmarking *SEKA* against baselines and exploring the impact of different $\delta_{\min}$ for selecting KV heads.

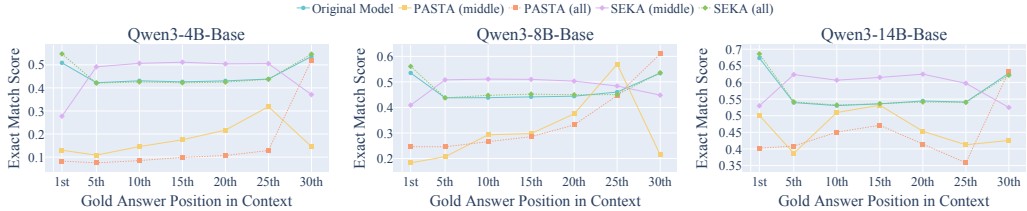

Figure 4: Exact match scores on the lost-in-the-middle task for Qwen3 models of three different sizes, comparing the original model, PASTA/*SEKA* applied to the middle region ($5^{th}$ to $25^{th}$ passages), and PASTA/*SEKA* applied to all passages.

***SEKA* Can Invert the U-shape Performance.** The results, summarised in Figure 4, reveal two primary findings. First, applying *SEKA* selectively to the middle passages (positions 5 to 25, which is a very rough range) is highly effective at inverting the canonical U-shaped performance profile: exact match scores at central positions substantially increase, eliminating the typical performance trough for answers located in the middle of long contexts. Second, applying *SEKA* uniformly across all passages can slightly exacerbate the lost-in-the-middle issue. The most noticeable improvements typically occur at the beginning or end positions, while enhancements in the middle are less pronounced or may even decrease. In contrast, PASTA is less effective for this task. Applying it to either the middle region or the entire context results in performance generally below the original baseline across all model sizes.

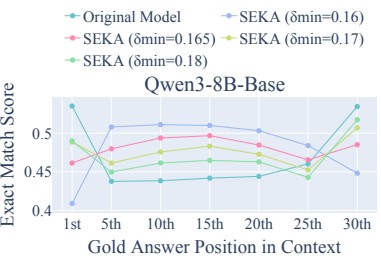

Figure 5: Exact match scores when applying *SEKA* to the middle region with different threshold $\delta_{\min}$.

***SEKA* Can Mitigate and Flatten the U-Shape When Applied to Appropriate Number of KV Heads.** In this control experiment, we fix the positive and negative steering gain coefficients ($g^+$ and $g^-$) at 0.2 and 0.1 respectively, and vary only the threshold $\delta_{\min}$ to control the number of steered KV heads. In practice, decreasing $\delta_{\min}$ increases the number of steered heads: for example, thresholds of 0.16, 0.165, 0.17, and 0.18 correspond to *SEKA* being applied on 58, 48, 41, and 31 KV heads for Qwen3-8B-Base, respectively. As shown in Figure 5, with an appropriate threshold $\delta_{\min}$ (around 0.165 and 0.17) and steering the middle region, *SEKA* can flatten the U-shaped performance curve without significantly compromising accuracy at the beginning and end positions. Note that the optimal threshold may vary with model size. Complete results for the 4B and 14B models are provided in Appendix K.

## 6 OVERHEAD ANALYSIS

A key advantage of our pre-computation approach is its compatibility with optimised mechanisms like FlashAttention (Dao et al., 2022; Dao, 2024; Shah et al., 2024). We quantify this by measuring inference overhead on 100 samples (avg. 4362 tokens) from Section 5.2 using a Qwen3-8B-Base model on a single NVIDIA-GH200-120GB GPU.

As shown in Table 3, the overhead for *SEKA* is negligible (+0.03s per sample). This efficiency is particularly notable as, for a fair comparison with PASTA, we use an aggressive configuration that steers 175 out of 288 available KV heads. In contrast, post-hoc methods incur significant costs. PASTA's reliance on editing the full attention matrix makes it incompatible with FlashAttention, leading to a substantial increase in latency (+1.03s) and memory usage (+23.12 GB). SPA, while memory-efficient for single samples, does not support batch processing and is thus the slowest overall. Our adaptive variant, *AdaSEKA*, introduces a moderate overhead for its dynamic, query-aware capabilities (+0.27s). However, it remains significantly more efficient than both PASTA and SPA, making it a far more practical option for steering in long-context scenarios.

Table 3: Inference overhead on Qwen3-8B-Base. Time is per-sample; memory is average peak usage.

| Method | Avg. Time (s) | Peak Mem. (GB, B=10) | Peak Mem. (GB, B=1) |
|---|---|---|---|
| Original | 0.55 | 27.63 | 16.72 |
| PASTA | 1.58 (+1.03) | 50.75 (+23.12) | - |
| SPA | 5.87 (+5.32) | - | 17.71 (+0.99) |
| *SEKA* | 0.58 (+0.03) | 27.66 (+0.03) | 16.75 (+0.03) |
| *AdaSEKA* | 0.82 (+0.27) | 43.22 (+15.59) | 18.23 (+1.51) |

## 7 RELATED WORK

Research on steering large language models falls into two main paradigms. Activation Steering (Dathathri et al., 2020; Subramani et al., 2022; Hernandez et al., 2024) guides high-level semantic outputs by intervening in MLP layers, while Attention Steering, the focus of our work, directs the model's focus to specific tokens within the input prompt.

**Activation Steering.** This line of work, also known as representation engineering, adds "steering vectors" to MLP layer activations to control semantic attributes (Zou et al., 2023). Applications include enhancing honesty and safety (Ravfogel et al., 2020; Burns et al., 2023; Iskander et al., 2023; Li et al., 2023a; Wei et al., 2023; Bhattacharjee et al., 2024; Qiu et al., 2024), controlling style (Turner et al., 2023; 2024), improving reasoning (Tang et al., 2025), and knowledge editing (Fang et al., 2025). Recent studies suggest these methods only work when the model already knows the target knowledge (Simhi et al., 2025). These methods are therefore different from our approach as they change what the model knows through its hidden states, but we control where the model looks via its attention mechanism.

**Attention Steering.** To address the challenge of LLMs failing to attend to key information in long contexts (Liu et al., 2024; Meng et al., 2022), prompt highlighting methods intervene post-hoc on either the attention scores (Zhang et al., 2024) or final logits (Tian & Zhang, 2025). However, these interventions often introduce significant latency; for instance, editing the full attention matrix is incompatible with modern optimisations like FlashAttention (Dao et al., 2022; Dao, 2024; Shah et al., 2024). This efficiency bottleneck motivates the need for pre-computation alternatives that can steer attention without sacrificing compatibility with optimised architectures.

## 8 CONCLUSION

In this paper, we introduced *SEKA* and its adaptive variant, *AdaSEKA*, a new class of training-free attention steering methods that operate by modifying key embeddings before the attention computation. This pre-attention approach overcomes the core efficiency limitations of prior work, ensuring full compatibility with optimised implementations. Our experiments confirm that both methods achieve state-of-the-art results on a range of standard benchmarks, with *AdaSEKA*'s query-adaptive routing demonstrating particularly strong performance. These gains are achieved with negligible overhead, making our work a practical step towards building more controllable and efficient LLMs for long-context applications.

## REPRODUCIBILITY STATEMENT

To ensure the reproducibility of our research, all necessary materials have been made publicly available on `https://github.com/waylonli/SEKA`. This repository includes: (1) the full source code for our proposed methods, *SEKA* and *AdaSEKA*; (2) detailed instructions for running all the experiments; (3) the pre-computed projection matrices used in our evaluations; and (4) the pre-processed versions of the datasets.

The original datasets used in our evaluation are publicly available and are cited in Section 4. Specifically, the BIASBIOS, COUNTERFACT, and "Lost in the Middle" datasets are all distributed under the MIT License. Details regarding the evaluation samples and metrics calculation are provided in Appendix E, while hyperparameters are specified in Appendix F.

## ACKNOWLEDGEMENTS

We thank the reviewers and the area chair for their valuable feedback. We also thank Yifu Qiu for constructive discussions related to this project. The authors acknowledge the use of resources provided by the Isambard-AI National AI Research Resource (AIRR). Isambard-AI is operated by the University of Bristol and is funded by the UK Government's Department for Science, Innovation and Technology (DSIT) via UK Research and Innovation; and the Science and Technology Facilities Council [ST/AIRR/I-A-I/1023] (McIntosh-Smith et al., 2024).

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

## A  SYNTHETIC DATASET FOR TOKEN-LEVEL RELEVANCE SUPERVISION

To supervise attention steering, we construct a synthetic dataset that enables precise control over token-level relevance. Each sample comprises two contexts $(C_1, C_2)$, each paired with a question and answer tuple $(Q_1, A_1$ and $Q_2, A_2)$. This structure allows us to define relevance by contrasting identical token spans across different query contexts.

Table 4: Constructed prompt triplets for both answer spans. Each group provides a neutral, positive, and negative variant based on question-context alignment.

| Group | Prompt |
|---|---|
| **Neutral** | Context: The portfolio manager allocates `capital` across equities and bonds. |
| **Positive** | Question: What does the portfolio manager allocate across equities and bonds? |
|  | Context: The portfolio manager allocates `capital` across equities and bonds. |
| **Negative** | Question: What does the climate model simulate? |
|  | Context: The portfolio manager allocates `capital` across equities and bonds. |
| **Neutral** | Context: The climate model simulates `sea-level rise` under different scenarios. |
| **Positive** | Question: What does the climate model simulate? |
|  | Context: The climate model simulates `sea-level rise` under different scenarios. |
| **Negative** | Question: What does the portfolio manager allocate across equities and bonds? |
|  | Context: The climate model simulates `sea-level rise` under different scenarios. |

With every pair of $(C, Q, A)$ triplets, as shown in Table 4, we can derive two supervision samples: one for the answer span "capital" in $C_1$, and another for the answer span "sea-level rise" in $C_2$.

Table 5: Synthetic data instance.

| | |
|---|---|
| **Context 1** ($C_1$) | The portfolio manager allocates capital across equities and bonds. |
| **Context 2** ($C_2$) | The climate model simulates sea-level rise under different scenarios. |
| **Question 1** ($Q_1$) | What does the portfolio manager allocate across equities and bonds? |
| **Answer 1** ($A_1$) | capital |
| **Question 2** ($Q_2$) | What does the climate model simulate? |
| **Answer 2** ($A_2$) | sea-level rise |

For each answer, we construct three variants: (1) a positive (relevant) prompt where the question and context are aligned (e.g., $Q_1$ for $C_1$, and $Q_2$ for $C_2$), (2) a negative (irrelevant) prompt where the question mismatches the context (e.g., $Q_1$ for $C_2$, and $Q_2$ for $C_1$), and (3) a neutral prompt containing only the context. This allows us to collect three classes of key embeddings for the answer spans within the context: $h^+$ for positive, $h^-$ for negative, and $h$ for neutral. In Figure 1, we empirically show that, for some key-value heads, different token spans exhibit a consistent shift in their key embeddings from negative to positive variants. This validates the construction and use of these relevance supervision signals.

**Practical construction details.** The synthetic dataset is lightweight to produce. We use a fixed template as shown in Table 5 and automatically prompt an GPT-4o to produce contrastive samples, using the prompt provided in Figure 6. This process requires no manual annotation. After collecting the generated samples, we convert them into JSON format for subsequent use.

## B    MULTI-EXPERT PROJECTION LEARNING SAMPLES FOR *AdaSEKA*

After constructing the synthetic dataset, we prepared three additional task-specific datasets, making a total of four, for multi-expert projection learning (Section 3.3). As shown in Table 6, each sample consists of a neutral and a positive prompt pair. For the Counterfact (Meng et al., 2022) dataset and the BiasBios (De-Arteaga et al., 2019) datasets, these pairs are collected from their respective training sets, following the original prompt templates outlined in Table 1. For each sample, we extract the key embeddings for the answer spans directly from the context. A distinct procedure is adopted for HotpotQA (Yang et al., 2018) to account for its multi-hop nature. The context is formed by concatenating all candidate paragraphs, and the key embeddings from all supporting facts are subsequently extracted and concatenated. Each expert projection is learned from a set of 200 randomly sampled instances from the training set for each task, using a fixed random seed of 42 to ensure reproducibility.

## C    GEOMETRIC INTUITION OF THE *SEKA* TRANSFORMATION

To provide geometric insight into the effect of *SEKA*'s key editing, consider the case where the projection matrix $\boldsymbol{P}$ is given by $\boldsymbol{U}\boldsymbol{U}^\top$, with $\boldsymbol{U} \in \mathbb{R}^{d_k \times r}$ having orthonormal columns that span the relevance subspace (i.e., $\boldsymbol{U} = \boldsymbol{U}^+$ or $\boldsymbol{U}^-$ as previously defined). For simplicity, assume $g = 1$ and focus solely on the positive (or negative) projection. The transformation then becomes:

$$\boldsymbol{k}'_j = (\boldsymbol{I} + \boldsymbol{U}\boldsymbol{U}^\top)\boldsymbol{k}_j. \tag{7}$$

Any vector $\boldsymbol{x} \in \mathbb{R}^{d_k}$ can be decomposed as

$$\boldsymbol{x} = \boldsymbol{x}_\| + \boldsymbol{x}_\perp, \quad \text{where} \quad \boldsymbol{x}_\| = \boldsymbol{U}\boldsymbol{U}^\top\boldsymbol{x}, \ \ \boldsymbol{x}_\perp = \boldsymbol{x} - \boldsymbol{U}\boldsymbol{U}^\top\boldsymbol{x}. \tag{8}$$

This decomposition is orthogonal. Specifically,

$$\boldsymbol{x}_\|^\top \boldsymbol{x}_\perp = (\boldsymbol{U}\boldsymbol{U}^\top\boldsymbol{x})^\top(\boldsymbol{x} - \boldsymbol{U}\boldsymbol{U}^\top\boldsymbol{x}) = \boldsymbol{x}^\top\boldsymbol{U}\boldsymbol{U}^\top\boldsymbol{x} - \boldsymbol{x}^\top\boldsymbol{U}\boldsymbol{U}^\top\boldsymbol{U}\boldsymbol{U}^\top\boldsymbol{x} \tag{9}$$

$$= \boldsymbol{x}^\top\boldsymbol{U}\boldsymbol{U}^\top\boldsymbol{x} - \boldsymbol{x}^\top\boldsymbol{U}\boldsymbol{U}^\top\boldsymbol{x} = 0, \tag{10}$$

using the idempotency of the projection ($(\boldsymbol{U}\boldsymbol{U}^\top)^2 = \boldsymbol{U}\boldsymbol{U}^\top$).

---

**Synthetic Samples Generation Prompt**

We are collaboratively generating a total of 100 synthetic examples.

You will generate examples in batches of exactly 20 per response. Across ALL batches in this conversation, every example must be globally unique.

Before generating a new batch, you MUST:
1. Review ALL previous examples in the conversation.
2. Ensure no repeated entities, contexts, events, sentence structures, questions, or answers.
3. Ensure no near-duplicates, paraphrased duplicates, or re-themed duplicates.

Each example must follow this structure (exact formatting):
Example k:
Context 1: `<C1>`
Context 2: `<C2>`
Question 1: `<Q1>`
Answer 1: `<A1>`
Question 2: `<Q2>`
Answer 2: `<A2>`

Generation requirements:
- Each context must be one concise fictional sentence.
- C1 and C2 must be semantically unrelated.
- Q1 must ask about a span that appears verbatim in C1.
- Q2 must ask about a span that appears verbatim in C2.
- A1 and A2 must be exact substrings (contiguous spans) of C1 and C2.
- No context, entity, theme, setting, or question type may repeat across any batch.
- Avoid any resemblance to earlier examples in wording, structure, or domain.
- No extra explanation or commentary.

Your task now: Read all previous examples in the conversation so far. Then generate the next 20 completely new, globally unique examples. Stop after exactly 20 examples.

---

Figure 6: Prompt template used to generate synthetic contrastive examples.

Applying the transformation, we have

$$(\boldsymbol{I} + \boldsymbol{U}\boldsymbol{U}^\top)\boldsymbol{x} \;=\; \boldsymbol{x} + \boldsymbol{U}\boldsymbol{U}^\top\boldsymbol{x} \;=\; \left(\boldsymbol{x} - \boldsymbol{U}\boldsymbol{U}^\top\boldsymbol{x}\right) + \boldsymbol{U}\boldsymbol{U}^\top\boldsymbol{x} + \boldsymbol{U}\boldsymbol{U}^\top\boldsymbol{x} \tag{11}$$

$$= \underbrace{\boldsymbol{x} - \boldsymbol{U}\boldsymbol{U}^\top\boldsymbol{x}}_{\boldsymbol{x}_\perp} + 2\underbrace{\boldsymbol{U}\boldsymbol{U}^\top\boldsymbol{x}}_{\boldsymbol{x}_\parallel} \;=\; \boldsymbol{x}_\perp + 2\boldsymbol{x}_\parallel. \tag{12}$$

which shows that the component along the subspace is amplified (doubled), while the orthogonal component remains unchanged.

While the $g = 1$ case offers geometric clarity, the result generalises for any $g \in \mathbb{R}$:

$$(\boldsymbol{I} + g\boldsymbol{U}\boldsymbol{U}^\top)\boldsymbol{x} = \boldsymbol{x}_\perp + (1 + g)\boldsymbol{x}_\parallel. \tag{13}$$

Thus, the relevance-aligned component is scaled by $(1 + g)$, while all orthogonal directions are preserved. This operation is neither a projection nor an orthogonal transformation, but a targeted linear modification that selectively amplifies directions aligned with the relevance subspace. *SEKA* leverages this property to boost relevant token features in a controlled and interpretable manner, enabling precise, token-wise attention steering without interfering with unrelated components.

While the geometric interpretation above clarifies how *SEKA* amplifies components of key vectors aligned with a learned relevance subspace, it is important to clarify what this subspace represents. *SEKA* is *not* intended to encode or manipulate semantic meaning. Its effect is deliberately confined

Table 6: Constructed prompt pairs for multi-expert projection learning. Each dataset provides neutral and positive variants based on question-context alignment.

| Dataset | Variant | Prompt |
|---|---|---|
| **CounterFact** | **Neutral** | Context: Previously, The mother tongue of Danielle Darrieux is French. Currently, The mother tongue of Danielle Darrieux is English. |
| | **Positive** | Question: Danielle Darrieux, a native ? Context: Previously, The mother tongue of Danielle Darrieux is French. Currently, The mother tongue of Danielle Darrieux is English. |
| **BiasBios** | **Neutral** | Context: Nora is an assistant professor of international relations at Bahcesehir University in Istanbul. She is also a Ronald D. Asmus Policy Entrepreneur Fellow with the German Marshall Fund and is a Visiting Fellow at the Centre for International Studies (CIS) at the University of Oxford. This commentary first appeared at Sada, an online journal published by the Carnegie Endowment for International Peace. |
| | **Positive** | Question: Nora has the occupation of a/an ? Context: Nora is an assistant professor of international relations at Bahcesehir University in Istanbul. She is also a Ronald D. Asmus Policy Entrepreneur Fellow with the German Marshall Fund and is a Visiting Fellow at the Centre for International Studies (CIS) at the University of Oxford. This commentary first appeared at Sada, an online journal published by the Carnegie Endowment for International Peace. |
| **HotpotQA** | **Neutral** | Context: Radio City (Indian radio station) Radio City is India's first private FM radio station and was started on 3 July 2001. . . . Arthur's Magazine Arthur's Magazine (1844–1846) was an American literary periodical published in Philadelphia in the 19th century. Edited by T.S. Arthur, it featured work by Edgar A. Poe, J.H. Ingraham, Sarah Josepha Hale, Thomas G. Spear, and others. In May 1846 it was merged into "Godey's Lady's Book". . . . First for Women First for Women is a womanś magazine published by Bauer Media Group in the USA. The magazine was started in 1989. It is based in Englewood Cliffs, New Jersey. . . . The company started first as a denim line, later evolving into a men's and women's clothing line. |
| | **Positive** | Question: Which magazine was started first Arthur's Magazine or First for Women? Context: Radio City (Indian radio station) Radio City is India's first private FM radio station and was started on 3 July 2001. . . . Arthur's Magazine Arthur's Magazine (1844–1846) was an American literary periodical published in Philadelphia in the 19th century. Edited by T.S. Arthur, it featured work by Edgar A. Poe, J.H. Ingraham, Sarah Josepha Hale, Thomas G. Spear, and others. In May 1846 it was merged into "Godey's Lady's Book". . . . First for Women First for Women is a woman's magazine published by Bauer Media Group in the USA. The magazine was started in 1989. It is based in Englewood Cliffs, New Jersey. . . . The company started first as a denim line, later evolving into a men's and women's clothing line. |

to the attention to route subspace of the transformer, consistent with prior mechanistic analyses (Elhage et al., 2021; Olsson et al., 2022).

Modern transformer-circuits work decomposes the action of an attention head as

$$H^{(h)}(R) = A^{(h)}(R) \otimes W_O^{(h)} W_V^{(h)} R, \qquad (14)$$

where $A^{(h)}$ is the query-key similarity tensor governing which tokens attend to which, and $W_O^{(h)} W_V^{(h)}$ writes attended features into the residual stream (Elhage et al., 2021). This formulation explicitly separates *routing* (Q/K) from *semantic write operations* (V/MLP).

Further, studies of induction and retrieval heads (Olsson et al., 2022) show that Q/K vectors implement token-matching and algorithmic routing behaviour, such as copying and continuation, while semantic information is primarily stored in value vectors and MLP activations. These findings align with our design that *SEKA* aims to modify only the routing (relevance) subspace, leaving the semantic subspace untouched.

# D  *SEKA* AND *AdaSEKA* ALGORITHMS

We provide detailed pseudocode for our proposed methods, *SEKA* and *AdaSEKA*. Algorithm 1 details the standard *SEKA* method. It involves an offline phase to learn fixed positive and negative projection matrices from contrastive data using SVD. During inference, a hook then applies these static projections to the key embeddings of highlighted tokens. Algorithm 2 describes the more flexible *AdaSEKA* framework. In essence, standard *SEKA* can be viewed as a special case of *AdaSEKA* with a single expert and no dynamic coefficient calculation. *AdaSEKA* generalises this by loading a bank of multiple expert SVD components offline. For each new prompt, it then performs a dynamic, query-aware pre-computation: it calculates routing coefficients based on the query's alignment with each expert and constructs a bespoke projection matrix on-the-fly. This tailored projection is subsequently applied during generation via the key-editing hook.

---

**Algorithm 1** Spectral Editing Key Amplification (*SEKA*)

**Require:** Triplets $\{\boldsymbol{h}, \boldsymbol{h}^+, \boldsymbol{h}^-\}_{\ell,h}$, variance threshold $\gamma$, gains $g^+, g^-$
**Ensure:** Projections $\{\boldsymbol{P}_{\ell,h}^+, \boldsymbol{P}_{\ell,h}^-\}$ and a key-editing hook
 1: **for all** layer $\ell$ **and** head $h$ **do**
 2:     $\boldsymbol{\Omega}_{\ell,h}^+ \leftarrow \frac{1}{n}\boldsymbol{h}^\top\boldsymbol{h}^+, \quad \boldsymbol{\Omega}_{\ell,h}^- \leftarrow \frac{1}{n}\boldsymbol{h}^\top\boldsymbol{h}^-$
 3:     $(\boldsymbol{U}_{\ell,h}^+, \boldsymbol{S}_{\ell,h}^+, \boldsymbol{V}_{\ell,h}^+) \leftarrow \mathrm{SVD}(\boldsymbol{\Omega}_{\ell,h}^+), \; (\boldsymbol{U}_{\ell,h}^-, \boldsymbol{S}_{\ell,h}^-, \boldsymbol{V}_{\ell,h}^-) \leftarrow \mathrm{SVD}(\boldsymbol{\Omega}_{\ell,h}^-)$
 4:     $k^+ \leftarrow \min\{k : \sum_{i=1}^k \boldsymbol{S}_{\ell,h,i}^+ / \sum_i \boldsymbol{S}_{\ell,h,i}^+ \geq \gamma\}, \; k^- \leftarrow \min\{k : \sum_{i=1}^k \boldsymbol{S}_{\ell,h,i}^- / \sum_i \boldsymbol{S}_{\ell,h,i}^- \geq \gamma\}$
 5:     $\boldsymbol{P}_{\ell,h}^+ \leftarrow \boldsymbol{U}_{\ell,h,:k^+}^+ \boldsymbol{U}_{\ell,h,:k^+}^{+\top}, \; \boldsymbol{P}_{\ell,h}^- \leftarrow \boldsymbol{U}_{\ell,h,k^-:}^- \boldsymbol{U}_{\ell,h,k^-:}^{-\top}$
 6: **end for**
 7: **Hook** applied to each selected $(\ell, h)$ *(registered per layer $\ell$; $\ell$ is fixed within the hook).*
 8:     **Input:** $K \in \mathbb{R}^{B \times T \times H \times d}$, mask $m$
 9:     **Note:** For brevity we omit the explicit layer index on $K$; projections remain $\boldsymbol{P}_{\ell,h}^{\pm}$.
10:     **for** $b=1..B, \; t=1..T, \; h=1..H$ **do**
11:         **if** $m_{b,t}=1$ **then**
12:             $\Delta \leftarrow \left(g^+ \boldsymbol{P}_{\ell,h}^+ + g^- \boldsymbol{P}_{\ell,h}^-\right) K[b,t,h,:]/2$
13:             $K[b,t,h,:] \leftarrow K[b,t,h,:] + \Delta$
14:     **return** $K$ to the attention computation
15: Register the hook for selected $(\ell, h)$ before generation and remove it afterwards.

---

# E  DETAILS OF STANDARD BENCHMARKS

We evaluate our method on three established benchmarks adapted from the PASTA framework (Zhang et al., 2024). We introduce significant improvements to the evaluation protocols, such as case-insensitive scoring, to ensure a more robust assessment. The JSON Formatting task was omitted as modern models achieve near-perfect performance, rendering it less useful for discriminating capabilities.

## E.1  COUNTERFACT

The COUNTERFACT benchmark (Meng et al., 2022) evaluates an LLM's ability to prioritise new contextual information over its pre-trained knowledge. Here, each fact is represented as a subject–relation–object triple $(s, r, o)$, where $s$ denotes the subject entity, $r$ the relation, and $o$ the object.

**Task Format.** The model receives input structured as: "Previously, $\{s \; r \; o_{\text{old}}\}$. Currently, $\{s \; r \; o_{\text{new}}\}$. $\{$question$\}$." The challenge arises because models often default to pre-trained associations rather than attending to the new, contradictory information provided in the context.

---

**Algorithm 2** Query-Driven Adaptive SEKA (*AdaSEKA*)

---

**Require:** SVD components $\{U_{m,\ell,h}^+, S_{m,\ell,h}^+\}$ for $M$ experts, top components $K$, gain $g$
**Ensure:** A key-editing hook using dynamically computed projections
 1: Store expert SVD components $\{U_{m,\ell,h}^+, S_{m,\ell,h}^+\}$ for all experts $m$, layers $\ell$, and heads $h$.
 2: **For a given prompt** with input IDs $I$:
 3:     Obtain last-token query vectors $q_{\ell,h}$ for each selected layer $\ell$ and head $h$.
 4: **for all** selected layer $\ell$ **and** head $h$ **do**
 5:     **for all** expert $m = 1..M$ **do**
 6:         Calculate coefficient $\alpha_{m,\ell,h}(q_{\ell,h}) \propto \sum_{k=1}^{K}(q_{\ell,h}^\top u_{m,\ell,h}^{+(k)}) \cdot \sigma_{m,\ell,h}^{+(k)}$ *(as per Eq. 6)*
 7:     **end for**
 8:     Construct $P_{\text{dynamic},\ell,h} \leftarrow \sum_{m=1}^{M} \alpha_{m,\ell,h}(q_{\ell,h}) \, U_{m,\ell,h,:,:K}^+ (U_{m,\ell,h,:,:K}^+)^\top$
 9:     Store $P_{\text{dynamic},\ell,h}$ for use in the hook.
10: **end for**
11: **Hook** applied to each selected $(\ell, h)$ *(registered per layer $\ell$; $\ell$ is fixed within the hook)*.
12:     **Input:** $K \in \mathbb{R}^{B \times T \times H \times d}$, mask $m$
13:     **Note:** For brevity we omit the explicit layer index on $K$.
14:     **for** $b=1..B$, $t=1..T$, $h=1..H$ **do**
15:         **if** $m_{b,t}=1$ **then**
16:             $\Delta \leftarrow g \cdot P_{\text{dynamic},\ell,h} K[b,t,h,:]$
17:             $K[b,t,h,:] \leftarrow K[b,t,h,:] + \Delta$
18:     **return** $K$ to the attention computation
19: Register the hook for selected $(\ell, h)$ before generation and remove it afterwards.

---

> **Prompt:** *"Previously, Kevin Garnett is a professional basketball player. Currently, \*\*Kevin Garnett is a professional baseball player\*\*. Kevin Garnett is a professional ___"*
>
> ........................................................................................................................................
>
> **Target:** *The model should generate "baseball player" rather than its pre-trained association of "basketball player".*

**Evaluation Metrics.**   Following (Zhang et al., 2024), to evaluate the model's ability to recall the new fact, we measure its internal preferences at the point of generation, rather than relying on parsing free-form text. For a given prompt, we provide the model with the entire context and question, and then assess the log probabilities it assigns to the potential next tokens.

- **Efficacy Score (ES):** This metric directly measures if the model prioritises the new, correct fact ($o_{\text{new}}$) over the old, incorrect fact ($o_{\text{old}}$). It is the percentage of times the model assigns a higher probability to the first token of the new fact than to the first token of the old fact. A high ES indicates that the model has successfully updated its belief based on the context.

$$\text{ES} = \frac{1}{N}\sum_{i=1}^{N} \mathbb{I}[P_{\text{LLM}}(o_{\text{new}}^{(i)}) > P_{\text{LLM}}(o_{\text{old}}^{(i)})]$$

- **Paraphrase Score (PS):** This metric measures generalisation by calculating the average Efficacy Score across a collection of human-written paraphrases of the original question.

### E.2   BIASBIOS

The BIASBIOS dataset (De-Arteaga et al., 2019) consists of biographies and was originally designed to explore gender bias in occupation prediction. The first sentence of each biography explicitly states the person's occupation, while subsequent sentences provide potentially distracting career details.

**Task Format.**   Each biography is appended with the prompt "{person} has the occupation of ___", and the model must predict the correct occupation from a list of 28 candidates.

> **Prompt:** *"\*\*Winnie is an American photographer living in New York.\*\* Specialized in fashion photography and portrait, she applies her talent on both humans and animals... Winnie has the occupation of"*
>
> -------------------------------------------------------------------------------------------------
>
> **Target:** *"photographer"*

**Evaluation Metrics.** We measure standard top-1 **Accuracy** across the 28 candidate occupations, implementing case-insensitive matching to ensure semantic equivalence is correctly evaluated.

### E.3 PRONOUNS CHANGING

This task evaluates instruction-following through linguistic transformation. Models are instructed to "*substitute 'she' and 'he' with 'they'*." This requires simultaneously adhering to the transformation rule while preserving the original content.

> **Prompt:** *"Mary is an Associate Professor in the Department of Curriculum Instruction at St. John University, she holds a doctorate in Reading/Writing/Literacy from the University of Pennsylvania... \*\*substitute 'she' and 'he' with 'they' and generate the occupation of Mary after changing pronouns\*\*."*
>
> -------------------------------------------------------------------------------------------------
>
> **Target:** *"Mary is an associate professor... they hold a doctorate... Mary has the occupation of Associate Professor."*

**Enhanced Evaluation Metric.** As noted during the public peer review of Zhang et al. (2024)[2], the original metric rewards empty strings for perfectly "converting" zero pronouns, resulting in misleadingly high scores. To address this, we introduce the **Pronoun-weighted Lexical Overlap Score (P. Score)**, which unifies instruction-following and content preservation into a single metric.

The P. Score modulates the credit for lexical overlap with the original text by the success rate of pronoun conversion. It is defined as:

$$\text{P. Score} = \frac{w_{\text{pron}} \times |T_{\text{ori}} \cap T_{\text{gen}}|}{|T_{\text{ori}}|}, \tag{15}$$

where $w_{\text{pron}}$ is the fraction of successfully converted pronouns, and $T_{\text{ori}}$ and $T_{\text{gen}}$ are the sets of non-pronoun content tokens from the original and generated texts, respectively. This ensures that empty generations receive a score of zero and that content preservation is only credited when instruction-following occurs. We evaluate two variants: one (P. Score) targeting core subject pronouns ("she", "he") and another (A. P. Score) targeting a complete set of gendered pronouns ("she", "he", "her", "him", "hers", "his", "herself", "himself").

## F TECHNICAL SETUP

This appendix section details the hyperparameters used for the *SEKA* and *AdaSEKA* experiments. For the CounterFact and Bias in Bios benchmarks, we performed a grid search to tune the hyperparameters on a validation set of 500 samples (indices 4500–4999), following the experimental setup of PASTA (Zhang et al., 2024). The final evaluation was then conducted on the test set (indices 5000–10000). For the Pronoun Changing task, hyperparameters were tuned on a separate small development set. All experiments across all models used greedy decoding.

The standard *SEKA* method requires tuning four hyperparameters: the variance threshold for projection construction ($\gamma$), the relevance-sensitivity threshold for KV-head selection ($\delta_{\text{min}}$) and the positive/negative steering gains ($g^+$ and $g^-$). The *AdaSEKA* framework simplifies this process, requiring only the tuning of the KV-head selection threshold ($\delta_{\text{min}}$) and a single steering gain coefficient ($g$). The selected hyperparameters for each model and task are provided in Table 7.

-------------

[2]https://openreview.net/forum?id=xZDWO0oejD&noteId=3kDI7QRqSI

Table 7: Hyperparameters for *SEKA* and *AdaSEKA* methods. *SEKA* uses the variance threshold ($\gamma$), KV-head selection threshold ($\delta_{\min}$), positive gain ($g^+$), and negative gain ($g^-$). *AdaSEKA* uses the KV-head selection threshold ($\delta_{\min}$) and steering gain ($g$).

| Model | Task | SEKA | | | | AdaSEKA | |
| --- | --- | --- | --- | --- | --- | --- | --- |
| | | $\gamma$ | $\delta_{\min}$ | $g^+$ | $g^-$ | $\delta_{\min}$ | $g$ |
| Qwen3-4B-Base | CounterFact | 0.960 | 0.13 | 1.56 | 0.00 | 0.1 | 3.0 |
| | Bias in Bios | 0.998 | 0.12 | 1.00 | 0.80 | 0.1 | 0.5 |
| | Pronoun Changing | 0.880 | 0.22 | 0.16 | 0.00 | 0.5 | 0.6 |
| Qwen3-8B-Base | CounterFact | 0.850 | 0.12 | 2.40 | 0.00 | 0.1 | 3.0 |
| | Bias in Bios | 0.998 | 0.12 | 0.60 | 0.30 | 0.1 | 0.5 |
| | Pronoun Changing | 0.900 | 0.20 | 0.19 | 0.00 | 0.5 | 0.6 |
| Qwen3-14B-Base | CounterFact | 0.870 | 0.10 | 2.42 | 0.00 | 0.1 | 3.0 |
| | Bias in Bios | 0.990 | 0.15 | 0.60 | 0.30 | 0.3 | 1.0 |
| | Pronoun Changing | 0.880 | 0.23 | 0.16 | 0.00 | 0.6 | 0.6 |
| Gemma-3-4B | CounterFact | 0.990 | 0.60 | 2.00 | 0.00 | 0.2 | 3.0 |
| | Bias in Bios | 0.800 | 0.12 | 0.80 | 0.00 | 0.2 | 0.8 |
| | Pronoun Changing | 0.800 | 0.20 | 0.40 | 0.00 | 0.4 | 1.0 |
| Gemma-3-12B | CounterFact | 0.990 | 0.50 | 1.00 | 0.00 | 0.1 | -5.0 |
| | Bias in Bios | 0.994 | 0.00 | 0.40 | 0.00 | 0.7 | 0.5 |
| | Pronoun Changing | 0.700 | 0.40 | -0.50 | 0.00 | 0.5 | -0.4 |

**Hyper-parameters Sensitivity.** To explore *SEKA*'s sensitivity to its hyper-parameters, we conduct an experimental analysis by varying each parameter independently while keeping all others fixed at their optimal configurations on the validation set (Table 7). We randomly select 500 test samples across the three benchmark tasks and adapt a one-at-a-time sweep over the following ranges: $\gamma \in \{0.75, 0.80, 0.85, 0.90, 0.95\}$, $\delta_{\min} \in \{0.10, 0.20, 0.30, 0.40, 0.50, 0.60\}$, $g^+ \in \{0.1, 0.2, 0.4, 0.6, 0.8, 1.0, 1.5, 2.0\}$, and $g^- \in \{0.00, 0.20, 0.40, 0.60, 0.80\}$.

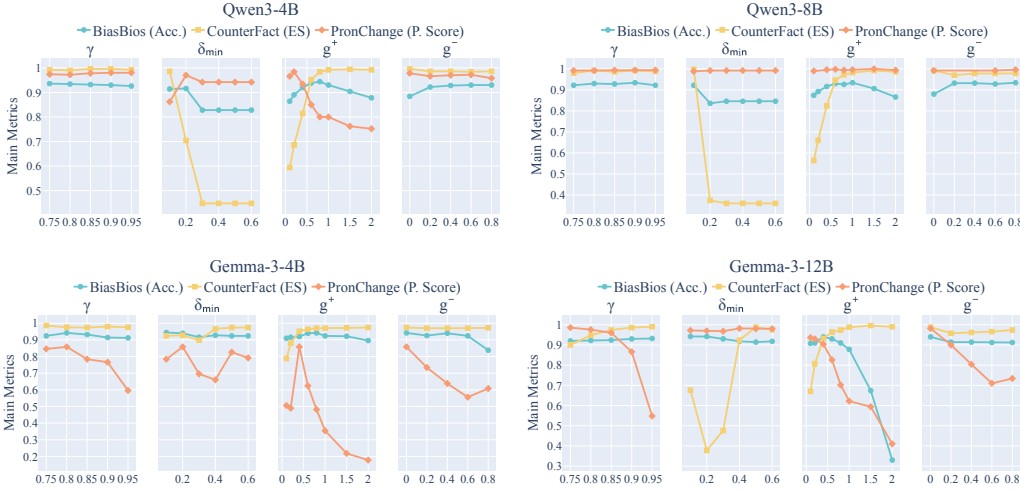

Figure 7: Sensitivity of *SEKA* to hyper-parameters across three benchmark tasks. Each curve varies a single hyper-parameter while keeping others fixed at their optimal settings on the validation set.

Three findings are observed from the results in Figure 7:

- $\delta_{\min}$ **and** $g^+$ **are the most influential.** These parameters determine which heads are steered and the strength of amplification. Performance drops when too few or too many heads

(depending on the tasks) are selected or when the positive gain is either too small to steer effectively or too large, which leads to over-amplification and degradation.

- **Models from the same family show similar trends.** Qwen3-4B and Qwen3-8B display nearly identical sensitivity patterns on CounterFact, both favouring low $\delta_{\min}$ and showing stability across $\gamma$. Gemma 3 models exhibit higher variance with respect to $\gamma$.

- **Task characteristics differ across models.** Stability patterns are task-model dependent. For example, Gemma-3-4B shows pronounced variability on PronChange at higher $g^+$ values, whereas CounterFact remains comparatively stable. In contrast, both Qwen3 models maintain strong robustness on BiasBios and PronChange but are noticeably more sensitive on CounterFact. These differences suggest that tasks requiring factual override (CounterFact) and tasks requiring instruction-following (PronChange) stress models in different ways, resulting in varying sensitivity.

## G  MECHANISTIC INSIGHT VIA ATTENTION VISUALISATION

To illustrate *SEKA* 's effect on model behaviour, we visualise the mean attention across all heads in selected layers for a CounterFact data sample: *"Previously Patrick Roy professionally plays the sport hockey. Currently Patrick Roy \*\*professionally plays the sport basketball\*\*. Patrick Roy is a professional ___"*. As shown in Figure 8, before *SEKA* is applied, the model's attention to the manipulated subspan ("was employed in Oslo") is low, with little focus on the relevant passage. After *SEKA* steering, attention in the affected layers becomes more concentrated on the target subspan, clearly demonstrating *SEKA* 's ability to selectively and effectively redirect model attention. This targeted effect aligns with the observed accuracy gains on benchmark tasks.

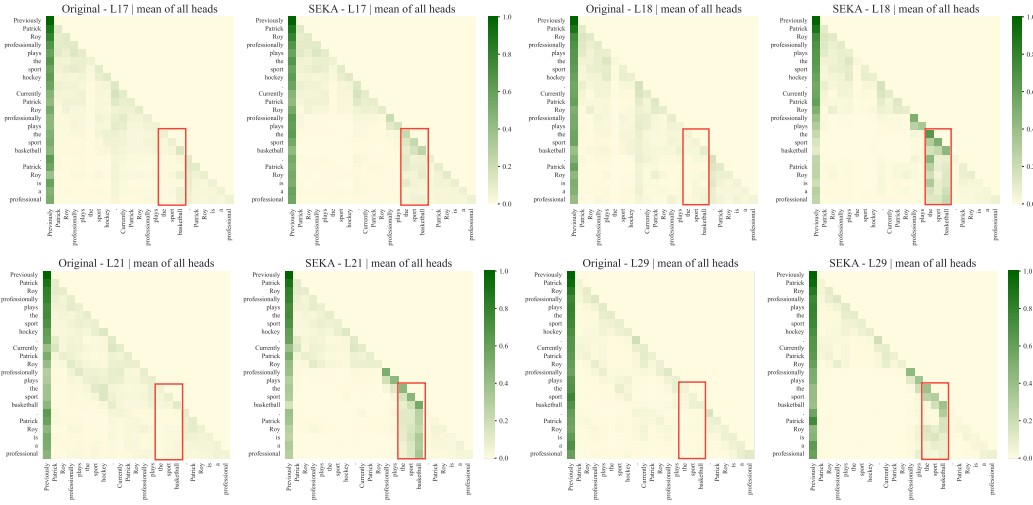

Figure 8: Layer-wise mean attention (all heads) in Qwen3-4B-Base at selected layers for the CounterFact data sample, shown before and after *SEKA* is applied.

## H  COMPLETE RESULTS OF PASTA WITH DIFFERENT CONFIGURATIONS

In the main results (Table 2), we reported the strongest performance for the PASTA baseline to ensure a fair comparison. For completeness, Table 8 provides a detailed breakdown of PASTA's performance across three different head-selection configurations. The first configuration replicates the original head search method, which identifies the top-$k$ performing heads by individually evaluating the steering effect of every attention head (Zhang et al., 2024). The other two configurations explore a hybrid approach by combining *SEKA*-style head selection with PASTA's attention steering. To address the misalignment between *SEKA*'s key-value head selection and PASTA's attention head steering, we test two strategies. The first is applying the *SEKA* selection computation directly on the outputs of the attention heads. The second uses the results of the key-value head selection and

applies them to attention heads via an interleaved repetition as the grouped-query attention mechanisms. For both hybrid methods, the selection criterion follows the *SEKA* methodology.

Table 8: Complete PASTA results with different configurations: (1) using *SEKA*'s KV-head configuration (runtime 1–2 minutes), (2) using attention head configuration transformed from *SEKA*'s KV-heads (1–2 minutes), and (3) using PASTA's original head-search routine (≈ 2 hours).

| Model | PASTA Configuration | CounterFact | | Bias in Bios | Pronoun Changing | |
|---|---|---|---|---|---|---|
| | | ES | PS | Acc. | P. Score | A. P. Score |
| Qwen3-4B-Base | *SEKA* KV-heads | 83.62 | 80.43 | 89.58 | 55.40 | 54.08 |
| | Transformed attention heads | 82.60 | 83.02 | 79.34 | 95.82 | 94.64 |
| | Original head-search | 97.16 | 96.03 | 87.84 | 47.20 | 47.20 |
| Qwen3-8B-Base | *SEKA* KV-heads | 82.08 | 71.72 | 86.32 | 62.91 | 61.44 |
| | Transformed attention heads | 78.20 | 79.69 | 77.30 | 98.86 | 98.72 |
| | Original head-search | 92.70 | 91.68 | 81.04 | 82.27 | 86.35 |
| Qwen3-14B-Base | *SEKA* KV-heads | 69.52 | 63.31 | 85.52 | 85.53 | 85.53 |
| | Transformed attention heads | 50.06 | 61.01 | 74.98 | 78.79 | 84.57 |
| | Original head-search | 76.84 | 66.33 | 88.46 | 90.98 | 90.94 |
| Gemma-3-4B | *SEKA* KV-heads | 55.66 | 37.05 | 71.42 | 67.39 | 66.43 |
| | Transformed attention heads | 78.36 | 59.53 | 82.58 | 21.89 | 21.65 |
| | Original head-search | 74.28 | 52.96 | 78.02 | 16.09 | 15.99 |
| Gemma-3-12B | *SEKA* KV-heads | 68.30 | 65.76 | 94.72 | 65.45 | 65.13 |
| | Transformed attention heads | 68.30 | 65.76 | 94.72 | 65.67 | 65.25 |
| | Original head-search | 62.68 | 71.72 | 76.24 | 68.47 | 68.01 |

# I   QUALITATIVE EXAMPLES

## I.1   COUNTERFACT EXAMPLES

Examples in Table 9 and 10 illustrate *SEKA*'s ability to steer the model towards newly provided factual information in the prompt, overriding its pre-trained knowledge.

Table 9: CounterFact Example 1: Overriding a known location.

| Prompt |
|---|
| Previously David Sainsbury, Baron Sainsbury of Turville used to work in London. **Currently David Sainsbury, Baron Sainsbury of Turville used to work in Berlin.** After 15 years of work, he published Pillboxes in 1985. David Sainsbury, Baron Sainsbury of Turville took up work in ___ |

| Model | Generation | Correct? |
|---|---|---|
| Gemma3-4B | London | ✗ |
| *SEKA*-Gemma3-4B | Berlin | ✓ |

Table 10: CounterFact Example 2: Overriding a known job title.

| Prompt |
|---|
| Previously Jean Baptiste Pompallier holds the position of bishop. **Currently Jean Baptiste Pompallier holds the position of cardinal.** Jean Baptiste Pompallier has the position of ___ |

| Model | Generation | Correct? |
|---|---|---|
| Gemma3-4B | bishop | ✗ |
| *SEKA*-Gemma3-4B | cardinal | ✓ |

## I.2 Bias in Bios Examples

Examples shown in Table 11 and 12 demonstrate *SEKA*'s ability to focus the model's attention on the correct, highlighted sentence in a biography, ignoring distracting information.

Table 11: Bias in Bios Example 1: Correcting a prediction.

| Prompt |
| --- |
| Tania **is an Illinois poet who has published one chapbook... and two full-length collections**. Her most recent book... is divided into ten sections... Its predecessor Simple Weight is also concerned with the Biblical narrative... Barbara Crooker said of that collection, "The poems have weight emotional, spiritual, political but are anything but simple.".Tania has the occupation of a/an ___ |

| | Qwen3-4B (Original) | *SEKA*-Qwen3-4B |
| --- | --- | --- |
| **Top Prediction** | journalist | poet |
| **Target** | poet | poet |

Table 12: Bias in Bios Example 2: Improving confidence in the correct answer.

| Prompt |
| --- |
| Michelle, PhD **is an Adjunct Assistant Professor in the Emory Department of Psychiatry and Behavioral Sciences**. She received her Ph.D. in 2003 from Emory University. Her areas of expertise include intergenerational processes of risk and resilience... She supervises postdoctoral residents... Michelle has the occupation of a/an ___ |

| | Qwen3-4B (Original) | *SEKA*-Qwen3-4B |
| --- | --- | --- |
| **Top Prediction** | psychologist | psychologist |
| **Target** | professor | professor |
| **LogProb of Target ("professor")** | -16.45 | -14.28 (**Improved**) |

## I.3 Pronouns Changing Examples

Tables 13 and 14 demonstrate how *SEKA* addresses specific failures in following instructions. While the original models generally perform well in this task (over 0.9 P. Score), they occasionally make errors. *SEKA* can improve performance in certain instances.

Table 13: Pronouns Changing Example 1.

| Prompt |
| --- |
| Jason is an assistant professor of English... **For the aforementioned text, substitute 'she' and 'he' with 'they' and generate the occupation of Jason after changing pronouns.** |

| **Qwen3-4B Generation (Incorrect)** |
| --- |
| *Jason is an assistant professor of English and as well as the coordinator of digital writing at Miami University of Ohio. He received his PhD from Ohio State University in English with a focus on rhetoric, composition, and literacy. His most current publication is "Re-Inventing Invention: A Performance in Three Acts." He is also a member of the Digital Writing Collaborative and a member of the Computers, Research, and Pedagogy committee.* |

| ***SEKA*-Qwen3-4B Generation (Corrected)** |
| --- |
| *Jason is an assistant professor of English and as well as the coordinator of digital writing at Miami University of Ohio. They received their PhD from Ohio State University in English with a focus on rhetoric, composition, and literacy. Their most current publication is "Re-Inventing Invention: A Performance in Three Acts." They are also a member of the Digital Writing Collaborative and a member of the Computers, Research, and Pedagogy committee.* |

Table 14: Pronouns Changing Example 2.

| Prompt |
| --- |
| Anna, JD, MA, LPC, is a depth psychotherapist in Houston, Texas... She holds degrees in philosophy, law, and clinical psychology... **For the aforementioned text, substitute 'she' and 'he' with 'they' and generate the occupation of Anna after changing pronouns.** |

| Qwen3-4B Generation (Incorrect) |
| --- |
| *Anna, JD, MA, LPC, is a depth psychotherapist in Houston, Texas, with emphasis and training in Jungian psychology.* She *holds degrees in philosophy, law, and clinical psychology.* She *is a frequent lecturer/teacher at The Jung Center, recently teaching their two-year extensive introduction to Jung and analytical psychology.* |

| *SEKA*-Qwen3-4B Generation (Corrected) |
| --- |
| *Anna, JD, MA, LPC, is a depth psychotherapist in Houston, Texas, with emphasis and training in Jungian psychology.* They *hold degrees in philosophy, law, and clinical psychology.* They *are a frequent lecturer/teacher at The Jung Center, recently teaching their two-year extensive introduction to Jung and analytical psychology.* |

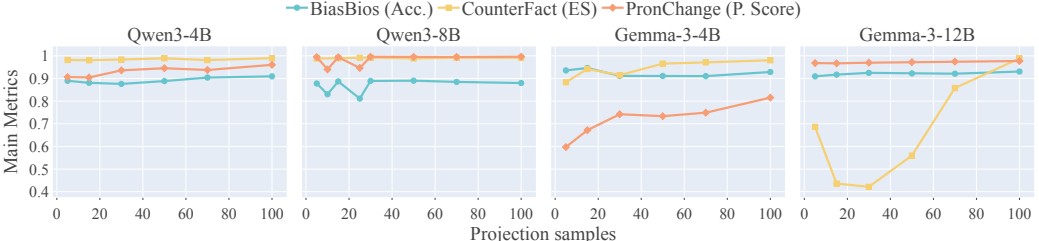

Figure 9: The performance of *SEKA* with varying numbers of synthetic samples used for learning projections across different models and tasks.

## J    PROJECTION SAMPLE EFFICIENCY ANALYSIS

To explore how varying data quantity of synthetic samples affect the quality of learned subspace representations, we conduct an analysis on the end-to-end performance on the three tasks in the standard benchmark using projections extracted from different number of synthetic samples for *SEKA*.

As shown in Figure 9, *SEKA* is generally data efficient across models and tasks. Performance typically stabilises once roughly 50 synthetic samples are used, though the exact threshold depends on the task, architecture, and model size.

More samples do not always yield higher peak performance, but they consistently produce more stable behaviour. With only a few samples, projections can overfit to the synthetic pairs and introduce unpredictable variance. Larger sample sizes mainly reduce this variance even when accuracy plateaus.

Two additional observations emerge when breaking down the results. First, models within the same family display similar behaviour patterns. For Qwen3 models, CounterFact stabilises relatively early, while Gemma3 models, especially Gemma3-12B, require more samples for the same task. BiasBios and Pronouns Changing tend to stabilise faster across most settings. Second, though family-level similarities are observed, model size still introduces noticeable differences. Qwen3-8B is the clearest example: both Pronoun Changing and BiasBios fluctuate when fewer than 50 samples are used but become stable afterwards, but this fluctuation is not observed in Qwen3-4B.

# K  COMPLETE RESULTS FOR $\delta_{\min}$ THRESHOLD ON LOST-IN-THE-MIDDLE

As noted in Section 5.2, the optimal KV-head selection threshold ($\delta_{\min}$) can vary with model size. Figure 10 illustrates the effect of varying this threshold on the performance of the Qwen3-4B and Qwen3-14B models.

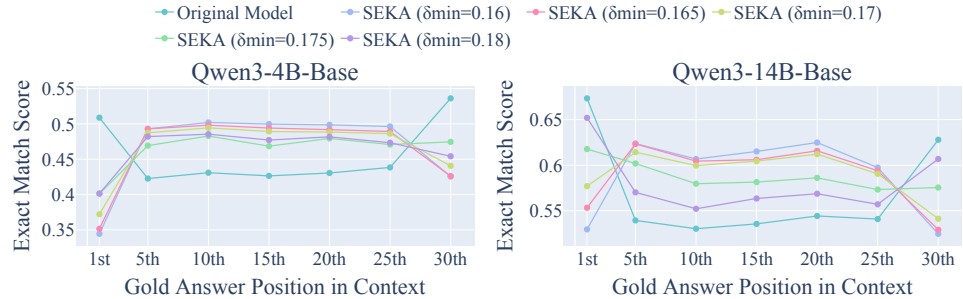

Figure 10: Exact match scores on the lost-in-the-middle task when applying *SEKA* to the middle region with different $\delta_{\min}$ thresholds for Qwen3-4B and Qwen3-14B.

# L  THE USE OF LARGE LANGUAGE MODELS (LLMS)

We used LLMs as general-purpose tools to refine the writing and debug the code for this paper. The LLMs were not used for research ideation or to generate any significant portion of the text. The authors take full responsibility for the content of this paper.

