# OpenReview forum: "Spectral Attention Steering for Prompt Highlighting"
_ICLR.cc/2026/Conference — ICLR 2026 Poster_

### Official Review · Reviewer_JuJy · 2025-10-25

**Soundness:** 3
**Presentation:** 2
**Contribution:** 3
**Rating:** 6
**Confidence:** 4

**Summary:**

The paper introduces SEKA, a new attention-steering approach designed for prompt highlighting. Motivated by the observation that most existing attention-steering methods operate directly on the attention matrix, which limits compatibility with efficient inference mechanisms such as FlashAttention, the authors propose modifying the key embeddings instead, prior to attention computation. Experiments demonstrate that SEKA and its adaptive variant, AdaSEKA, achieve promising performance on standard benchmarks and effectively mitigate positional bias.

**Strengths:**

1. The motivation is well-founded. Previous attention-steering methods are relatively direct and often incompatible with mainstream acceleration frameworks, revealing a significant gap between steering effectiveness and efficient inference.

2. The proposed SEKA and its variant offer a simple yet effective approach. By modifying the key embeddings along directions associated with the highlighted context, the model can better emphasize relevant information.

3. The approach demonstrates strong empirical performance, achieving near-perfect scores on multiple benchmarks and effectively alleviating positional bias.

**Weaknesses:**

1. The construction of the relevance context appears tedious, and the overall pipeline is not clearly explained. The process of building positive and negative embeddings is a key component of both SEKA and AdaSEKA, yet it is not described in the main text, and the Appendix also lacks sufficient details. It remains unclear whether the creation of this synthetic dataset is required for each benchmark individually, and how the authors match pairs of context (C), question (Q), and answer (A), i.e., whether these are randomly sampled from the test set or selected according to specific criteria. Although SEKA enables efficient inference, the offline construction of such a synthetic dataset could potentially introduce significant preprocessing overhead.

2. The trade-off between the positive and negative directions in SEKA seems somewhat naive. The authors simply assign equal weights (0.5) to both directions, without further justification or analysis. It would be valuable to understand whether the positive embeddings play a more critical role in emphasizing relevant information, and whether asymmetric weighting might lead to better performance or stability.

3. The paper introduces a large number of hyperparameters, yet provides no ablation studies or sensitivity analyses to assess their impact. In particular, parameters such as γ (for singular vector selection) in SEKA, the number of experts (m) in AdaSEKA, and the head selection threshold (δₘᵢₙ) are not thoroughly discussed. Moreover, the authors do not specify the exact configurations of these hyperparameters in the experiments, making it difficult to evaluate the robustness and reproducibility of the results.

**Questions:**

Please see the weakness.

---

> ### Author Response · Authors · 2025-11-20
> **Reply to Reviewer JuJy**
>
> Thank you for the constructive review and for recognising the motivation and empirical strengths of SEKA and AdaSEKA. Following your comments, we added detailed explanations of the synthetic data construction, clarified task-specific sampling procedures, and conducted new hyper-parameter sensitivity analyses. Below we address each point in turn.
>
> > The construction of the relevance context appears tedious, and the overall pipeline is not clearly explained...
>
> Thank you for raising this point. We agree that these details should be clearly stated, **so we have added an explanation in Appendix A, lines 746-749, as well as the prompt template in Figure 5**. The construction of the synthetic dataset is in fact lightweight: we define a fixed template (Table 4) and prompt an LLM to generate contrastive samples automatically.
>
> For AdaSEKA’s task-specific projections, the procedure is similarly simple. We *randomly sample 200 examples from the training split* of each benchmark (not the test split, to avoid leakage, as mentioned in Appendix B) and extract the answer span directly from the provided context. **We’ve now added a sentence in Appendix B, lines 796-798**, clarifying that the 200 samples are randomly selected using a fixed random seed of 42 for each task. Thank you for bringing this to our attention again. This was an oversight on our part.
>
> Regarding offline overhead, although SEKA/AdaSEKA require a one-time preprocessing step, it is fully automated and requires no human annotation. Projection extraction and KV-head selection can be run in parallel and typically complete *within 1–2 minutes (as mentioned in Appendix I)* depending on model size. In practice, this overhead is minimal compared to other methods like PASTA or those that require training or finetuning.
>
> > ... The authors simply assign equal weights (0.5) to both directions, without further justification or analysis ...
>
> Thank you for raising this point. There might be misunderstanding here and we have clarified the paper as follows: SEKA does not fix the positive and negative directions to equal weights. The gains $g^{+}$ and $g^{-}$ are hyperparameters tuned via a simple grid search, as detailed in Appendix G. Due to space constraints, this was only briefly mentioned and referenced from the main paper when we introduce the motivation of AdaSEKA in section 3.3, and **we have now updated the draft in lines 160-161 to clarify**: “$g^+, g^-$ are two independently adjustable scalars controlling the positive and negative steering gains.”
>
> Regarding if the positive projections play an important role, the answer is yes. **We have conducted extra hyper-parameter analysis at the end of Appendix G and this is one of the findings** (details in reply to next question blow). This is precisely why, in AdaSEKA, when multiple expert projections are available, we retain only the positive projections and rely on the adaptive routing mechanism to weight them dynamically. This reduces manual configuration while prioritising the most informative relevance-aligned directions.
>
> > The paper introduces a large number of hyperparameters, yet provides no ablation studies or sensitivity analyses to assess their impact.
>
> Thank you for the helpful observation. In fact, SEKA introduces only four hyperparameters in total $g^{+}, g^{-}, \gamma, \delta_{\min}$. *The full configurations used in all experiments are already documented in Table 7 of Appendix G so that the experiments are fully reproducible.*
>
> Regarding hyperparameter sensitivity analysis, we completely agree that it adds value to our work. Thanks very much for raising this valuable point! As briefly mentioned in our reply to your previous comment **we ran an additional set of experiments and added the full results to the end of Appendix G, paragraph “Hyper-parameters Sensitivity”, and Figure 7 in the revised paper draft** (seems OpenReview does not allow figures in the rebuttal, so please refer to the updated draft).
>
> To summarise what we found:
>
> - Across the board, SEKA is fairly robust. Most parameters have wide stable regions and only a couple require light tuning.
> - $\delta_{\text{min}}$ and $g^{+}$ matter the most. Performance drops if we steer too few heads, too many heads, or if $g^{+}$ is either too weak or too strong.
> - Models from the same family behave very similarly. The exact optimal values shift with model size, but the overall patterns are consistent.
> - Task characteristics vary between models, meaning stability patterns depend on both the task and the model. For example, Qwen-3 is stable on PronChange, while Gemma-3 is more sensitive. Conversely, Gemma-3 is more stable than Qwen-3 on Counterfact.
>
> ---
>
> We hope these clarifications and additional analyses address your concerns and strengthen your overall assessment of the work. If any part would benefit from further detail, please let us know. Thank you again for the careful and thoughtful review.

---

> ### Comment · Reviewer_JuJy · 2025-11-27
>
> Thank you to the authors for the thoughtful responses. Most of my concerns have been resolved, so I have decided to keep my original score.

---

### Official Review · Reviewer_Z3wt · 2025-10-31

**Soundness:** 3
**Presentation:** 3
**Contribution:** 3
**Rating:** 6
**Confidence:** 2

**Summary:**

The paper proposes a training-free attention–steering method and a query-adaptive variant to highlight user-specified spans by intervening on the attention inputs rather than post-hoc editing attention matrices.

**Strengths:**

1. By editing keys before attention, the approach avoids material incompatibilities with FlashAttention‐style kernels and sidesteps full matrix storage required by prior post-hoc methods.
2. Offline spectral learning over contrastive triplets is training-free at run time and generalizes across model sizes and two model families.
3. Strong empirical results with ablations and overhead accounting.

**Weaknesses:**

1. The approach relies on constructing synthetic contrastive triplets and performing SVD computations for each layer and head, as well as for every expert in the AdaSEKA variant.
2. Relevance triplets are synthetically constructed, and the steering effectiveness may hinge on these triplets.
3. The method involves tuning several thresholds and gain parameters for each model and task using dev sets. However, the paper provides limited analysis of parameter sensitivity or stability, and the optimal value of $\delta_{\min}$ appears to vary with model size.
4. The evaluation is limited to highlight-friendly setups and short-form QA/rewriting.

**Questions:**

1. Does SEKA increase the probability of obeying unsafe highlighted instructions? Any mitigation via head filtering or gain caps?
2. How often “steering all passages” degrades the middle in long-context QA?
3. Beyond heatmaps and PCA shifts, can you show that edits causally increase attention mass or logit contribution from highlighted keys to answer tokens in specific heads and layers, and that those heads are sufficient via knock-out and knock-in?

---

> ### Author Response · Authors · 2025-11-20
> **Reply to Reviewer Z3wt (Part 1: weaknesses)**
>
> Thank you for the constructive review and for recognising the strengths of SEKA’s design, compatibility, and empirical performance. Following your comments, we added new hyper-parameter sensitivity analyses, expanded the discussion on synthetic triplet construction, and clarified several methodological details. Below we address each of your points in detail.
>
> **Weaknesses**
>
> > The approach relies on constructing synthetic contrastive triplets and performing SVD computations for each layer and head, as well as for every expert in the AdaSEKA variant.
>
> Thanks for pointing this out. You’re right, this does add a bit of overhead during the offline stage, but very lightweight. It mainly involves some prompting to generate synthetic samples for SEKA, or a bit of programming to automatically extract data from a task-specific dataset. This process is straightforward, following the examples we’ll release with the codebase. Some more details about the construction of the synthetic dataset (quite lightweight): we define a fixed template (Table 4) and prompt an LLM (gpt-4o) to generate contrastive samples automatically, and the expert projection extraction is also fully automated and requires no human annotation. **We have added more technical details about the prompts, model used, and random seed setting in Appendix A and B (Appendix A: Figure 5, lines 746-749; Appendix B: 796-798)**.
>
> Projection extraction and KV-head selection (for layer and head) run in parallel and typically complete *within 1–2 minutes* depending on model size. In practice, this overhead is minimal compared to other methods like PASTA or those that require training or finetuning.
>
> > Relevance triplets are synthetically constructed, and the steering effectiveness may hinge on these triplets.
>
> As introduced above, our synthetic data generation does not need to be complicated. And task-specific projections extracted from existing dataset is also effective from the results of AdaSEKA. Our empirical results in Table 2 also show that projections learned from synthetic contrastive prompts generalise broadly: vanilla SEKA uses only these universal projections but manages to improve performance across multiple unseen tasks without any task-specific tuning. In addition, AdaSEKA shows that automatically selecting and weighting several task-specific projections can sometimes provide further improvements, even though its main goal is to reduce the number of hyperparameters.
>
> > ... the paper provides limited analysis of parameter sensitivity or stability, and the optimal value of  appears to vary with model size.
>
> Thank you for raising this. You are absolutely right that we need a clearer analysis of how sensitive SEKA is to its hyper-parameters. Following your suggestion, **we ran an additional set of experiments and added the full results to Appendix G, paragraph “Hyper-parameters Sensitivity”, and Figure 7 in the revised paper** (seems OpenReview does not allow figures in the rebuttal, so please refer to the updated draft).
>
> To summarise what we found:
>
> - Across the board, SEKA is fairly robust. Most parameters have wide stable regions and only a couple require light tuning.
> - $\delta_{\text{min}}$ and $g^{+}$ matter the most. Performance drops if we steer too few heads, too many heads, or if $g^{+}$ is either too weak or too strong.
> - Models from the same family behave very similarly. The exact optimal values shift with model size, but the overall patterns are consistent.
> - Task characteristics differ across models, i.e. stability patterns depend on both the task and the model. For example, Qwen-3 is stable on PronChange, while Gemma-3 is more sensitive. On the other hand, Gemma-3 is more stable than Qwen-3 on Counterfact.
>
> > The evaluation is limited to highlight-friendly setups and short-form QA/rewriting.
>
> Thank you for pointing this out. Our choice of standard benchmarks follows prior attention-steering work such as PASTA [1]. But we fully agree that short-form QA and rewriting alone would be insufficient, *which is why we further include the Lost-in-the-Middle evaluation*. This task probes long-context retrieval (though it is no longer that “long” given the hugely expanded context windows for nowadays LLMs) under heavy distraction and therefore moves beyond “highlight-friendly” setups, testing whether SEKA can prioritise relevant spans in more challenging conditions.
>
> [1] Zhang, Qingru, et al. Tell Your Model Where to Attend: Post-hoc Attention Steering for LLMs. ICLR 2024.

---

> ### Author Response · Authors · 2025-11-20
> **Reply to Reviewer Z3wt (Part 2: questions)**
>
> **Questions**
>
> > Does SEKA increase the probability of obeying unsafe highlighted instructions? Any mitigation via head filtering or gain caps?
>
> Thank you for raising this important safety question. A full safety evaluation is outside the scope of this paper, but we see two reasons why SEKA does not fundamentally increase risk and also two realistic mitigation paths going forward.
>
> First, SEKA only adjusts routing in the Q/K space. It tells the model where to focus, but it doesn't change the semantic content in the value vectors or the MLP stack, where most safety-related behaviors actually reside. As a result, harmful activations can still be detected by the guardrail.
>
> Second, our implementation already includes practical relevance-based head filtering, which keep the effect only with the retrieval heads. Gain caps are also possible but that might depend on the practical tasks.
>
> Your question also points a very interesting direction. Perhaps we can combine attention steering (SEKA) with activation steering. SEKA can strengthen retrieval of highlighted information, while activation steering can enforce semantic safety constraints on the generation itself. Since MLP activations come after attention in each transformer block, they can effectively act as a safety “buffer” even when attention is locally boosted. This might open up a whole new direction for research.
>
> Thanks for your inspiring question again!
>
> > How often “steering all passages” degrades the middle in long-context QA?
>
> Thank you for the question. In our experiments, steering all passages does not meaningfully degrade performance in the middle region. As shown in Figure 3, the behaviour is quite consistent: applying SEKA on all passages mainly boosts the beginning (and occasionally the end), but the middle itself remains essentially stable.
>
> Where small drops do occur, they are extremely minor (typically well under 0.01) and sometimes it may slightly improve which can also be seen in Figure 3. Overall, we do not see a systematic degradation pattern in the middle when steering is applied to all passages.
>
> > Beyond heatmaps and PCA shifts, can you show that edits causally increase attention mass or logit contribution from highlighted keys to answer tokens in specific heads and layers, and that those heads are sufficient via knock-out and knock-in?
>
> We appreciate your suggestion and agree that head-level causal analysis is an interesting direction. But we also want to note that if we understand correctly, this serve as a valuable direction through the interpretability but do not affect our main claim.
>
> Our understanding is that you are asking for evidence that (i) SEKA quantitatively increases attention mass or logit contribution from highlighted context tokens to answer tokens in specific heads/layers, and that (ii) these heads are causally responsible for the behavioural effect via knock-out (disabling SEKA only in those heads) or knock-in (applying SEKA only in those heads).
>
> We think the first question is already supported by the heatmap visualisation of the attention scores matrix. For the second aspect, we are a bit unsure as we feel like the filtering mechanism somehow plays the role of knock-out and knock-in.
>
> We value your suggestion and would be grateful for clarification to ensure we address the intended analysis correctly. In particular, we would appreciate confirmation if you are seeking any specific type of analysis. We would be happy to try our best to conduct it if it’s manageable within the rebuttal period.
>
> ---
>
> We hope the clarifications and additional analyses on hyper-parameters have addressed your concerns and strengthened your confidence of our work. If you need any more details or further clarification, please let us know. Thank you again for your time and effort in reviewing our paper!

---

### Official Review · Reviewer_XAQm · 2025-10-31

**Soundness:** 2
**Presentation:** 3
**Contribution:** 4
**Rating:** 8
**Confidence:** 3

**Summary:**

This paper discussed an improved attention steering techniques that combines the existing post-hoc attention steering with spectral decomposition on the key embedding. The results show that SEKA, especially its variant AdaSEKA, outperforms the existing attention steering approaches, SPA and PASTA on five instruction following tasks for most of the models. Since the SEKA performs decomposition and steering on the key embedding instead of the full attention matrix, it is also compatible with modern attention implementation such as FlashAttention. This advantage gives SEKA better real-applicability than the existing methods like PASTA that relies on the full attention modifications.

**Strengths:**

1. It is a clever integration of attention steering and spectral editing. The implementation of the method is well presented in the paper and is easy to reproduce.
2. The SEKA is compatible with advanced attention implementation that does not compute the full attention matrix. As a result, the method also has lower computation overhead compared to PASTA and SPA that require full attention editing.
3. The adaptive expert projection routing reduce the overhead in fine-tuning steering hyperparameters.
4. The performance of SEKA and AdaSEKA is robust for most models on most tasks. The fact that the SEKA can help model fetch information presented in the middle of a long input context, reversing the U-shape performance (lost in the middle) of existing method, highlights this method can be used for long-document QA.

**Weaknesses:**

1. Generalizability to unseen instructionw? The paper showed their method's generalizability in CounterFact tasks through paraphrase scores that measure performance on human-rewritten versions of the original questions. However, does the projection learned on one task or a collection of tasks generalize to the unseen instruction types.
2. It is unclear what's the relationship between the amount of data used in finding "relevance subspace" for a given instruction and the performance of the SEKA method. If projection matrix is domain-specific, how many pairs of instruction following responses and non-following responses are needed for SEKA to be effective.
3. The benefits of learned projections do not seem to be significant on the Qwen3 family.

**Questions:**

1. In table 2, why does PASTA underperform the model's original performance on pronoun changing tasks for Qwen3-14B and Gemma3-12B?
2. pg 6, ln 281, given all tested models are base models, what does "empty response" mean?

---

> ### Author Response · Authors · 2025-11-20
> **Reply to Reviewer XAQm**
>
> Thank you for the positive and encouraging review, and for recognising the strengths of SEKA and AdaSEKA across compatibility, efficiency, and long-context performance. Following your comments, we added a new data-efficiency analysis, and clarified questions regarding the generalisability of task-agnostic projections and the “empty response” issue existing in previous research. Below we address each of your points in detail.
>
> **Weaknesses**
>
> > ... does the projection learned on one task or a collection of tasks generalize to the unseen instruction types.
>
> Thank you very much for the comments but we think this might be a misunderstanding of the quantitative results in Table 2. It already shows that vanilla SEKA, which only uses projections learned from this synthetic dataset and *receives no task-specific tuning*, generalises well to three diverse unseen tasks. This provides empirical support that the learned relevance subspace is sufficiently broad to transfer beyond the synthetic setting as it’s not task-specific as the ones used for AdaSEKA though it requires less configuration efforts and sometimes achieves better performance.
>
> > It is unclear what's the relationship between the amount of data used in finding "relevance subspace" for a given instruction and the performance of the SEKA method...
>
> Thank you for this valuable suggestion. We fully agree it is important and **have added a new analysis in Appendix K**. Before we present the extra analysis results, we would first like to clarify that, for SEKA, the projection matrix is not required to be domain-specific. As discussed in our previous response, the projections learned from our synthetic contrastive data already generalise well.
>
> In terms of the analysis, we gradually increase the number of synthetic samples used for projection extraction and evaluate the resulting end-to-end performance on all three tasks in the standard benchmark. The plots and discussion are in **Appendix K (and in Figure 9**; unfortunately OpenReview does not allow us to attach the figure). Below is a short summary of the findings:
>
> - SEKA is generally data-efficient across models and tasks. Performance stabilises once roughly 50 synthetic samples are used.
> - Increasing the sample size does not always produce higher peak performance, but it consistently yields more stable behaviour.
> - Models within the same family exhibit similar trends, though model size still introduces noticeable differences.
>
> Thank you again for this valuable suggestion as it helps us improve the clarity of the sample efficiency and make the research work more complete.
>
> > The benefits of learned projections do not seem to be significant on the Qwen3 family.
>
> The gains on Qwen3 are still substantial on the two harder benchmarks. For Counterfact, SEKA and AdaSEKA lift Qwen3-4B from 57.7 to 99.02, exceeding PASTA (97.16). For BiasBios, Qwen3-4B improves from 82.94 to 91.86, again surpassing PASTA (89.58). Similar trends hold for Qwen3-8B and Qwen3-14B. The only case with limited headroom is Pronoun Changing, where Qwen3-14B already performs near-perfectly, leaving little room for further gains.
>
> **Questions**
>
> > In table 2, why does PASTA underperform the model's original performance on pronoun changing tasks for Qwen3-14B and Gemma3-12B?
>
> A plausible explanation is that PronChange is already a saturated task for models such as Qwen3-14B and Gemma-3-12B. In such cases, hand-crafted post-hoc rescaling of the attention matrix, as performed by PASTA, may introduce unnecessary perturbations that degrade an already-good attention pattern. That said, we want to note that this is our interpretation rather than a definitive causal claim. For our experiments, we tuned PASTA according to the their original head-search procedure and also tested multiple head configurations, even including those filtered by our proposed method, and always reported the best-performing results as introduced in Appendix I.
>
> > pg 6, ln 281, given all tested models are base models, what does "empty response" mean?
>
> Thanks for pointing this out. A clarification regarding this can be found in Appendix F.3. To give a bit more context, as noted in the public peer review of the PASTA paper (https://openreview.net/forum?id=xZDWO0oejD&noteId=3kDI7QRqSI) in ICLR 2024, the original PASTA evaluation metric can incorrectly mark an empty string (“”) or a fully irrelevant answer that contains none of the pronouns “they”, “he”, or “she” as correct. This issue arises because the metric only checks for pronoun presence rather than the correctness of the transformation. Our Appendix F.3 provides further explanation and presents our improved metric that avoids this failure mode.
>
> ---
>
> We hope these clarifications and analyses fully resolve your concerns and further strengthen your confidence in the method. If any aspect would benefit from additional detail, we are happy to provide further clarification. Thanks again for your time and constructive feedback.

---

### Official Review · Reviewer_c6N4 · 2025-11-01

**Soundness:** 3
**Presentation:** 4
**Contribution:** 3
**Rating:** 6
**Confidence:** 3

**Summary:**

This paper introduces a novel, training-free attention steering method called SEKA and its adaptive variant AdaSEKA for prompt highlighting. The core innovation lies in shifting from the traditional paradigm of "post-editing" the attention matrix to directly editing key vectors before attention computation. It learns a universal "relevance subspace" via offline spectral decomposition and constructs projection matrices to amplify the features of highlighted tokens. Experiments demonstrate that the method outperforms existing approaches on multiple standard benchmarks while adding negligible latency and maintaining full compatibility with modern efficient attention mechanisms like Flash Attention. Its primary contribution is a new paradigm for efficient, precise, and general-purpose attention steering.

**Strengths:**

- High Originality. Achieves a paradigm breakthrough by shifting from "editing attention output" to "editing attention input." The concept is novel.
-  Exceptional Efficiency and Compatibility
    \par Near-zero latency overhead and full compatibility with Flash Attention represent a decisive advantage over methods like PASTA, showing great practical potential.
-  Comprehensive and Robust Experiments
    \par Validation across multiple tasks, models, and metrics consistently demonstrates the method's effectiveness and generality. Ablation studies thoroughly confirm the value of the core components.
- Advanced Adaptive Mechanism
    \par AdaSEKA's query-aware routing mechanism dynamically combines experts, reducing the need for manual hyperparameter tuning and improving the method's intelligence and applicability.

**Weaknesses:**

- While the geometric interpretation is intuitive, the paper would be further strengthened by a linguistic or semantic characterization of the learned relevance subspace, for instance, by analyzing the most affected tokens or nearest neighbors in embedding space. Suggestion: Incorporate semantic analysis of the projection directions (e.g., via nearest-neighbor word analysis of projected keys).
- The construction of experts in AdaSEKA relies on manual task division. Currently, experts are constructed based on different datasets, with no exploration of automatic clustering or dynamic expert generation.
- Significant performance fluctuations in certain tasks: For instance, in the Pronoun Changing task on Gemma-3-12B, AdaSEKA exhibits noticeable performance volatility. Suggestion: Analyze the impact of model architecture on the method and develop a more stable routing mechanism.
- It would be helpful to discuss the data efficiency of the offline spectral learning process, such as how projection stability changes with sample size. While the current approach utilizes specially constructed synthetic datasets, it lacks systematic analysis of how varying data quantities affect the quality of learned subspace representations. This gap introduces uncertainties in the reproducibility and generalizability of the method in practical applications.

**Questions:**

- The rationale behind data construction. SEKA relies on an offline-constructed synthetic dataset to learn a "universal" relevance subspace. How do the authors justify that this specific construction of contrastive prompts can capture a sufficiently broad notion of relevance to generalize to unseen, and potentially more complex, tasks?
- Interpretability of Projection Directions. Beyond the geometric interpretation, do the primary directions of the learned positive/negative projection matrices correspond to identifiable linguistic features? Have the authors attempted to analyze these directions, for example, by examining the nearest neighbors in the vocabulary space of key vectors that are most amplified/diminished?
- Synergy with Positional Bias Methods. Given that SEKA operates on Key vectors and methods like "Found-in-the-Middle" calibrate positional bias, do the authors believe that combining SEKA with explicit positional bias calibration methods could yield further performance gains? Is this a worthwhile direction for future work?
- Could the source code be made publicly available?

---

> ### Author Response · Authors · 2025-11-20
> **Reply to Reviewer c6N4 (Part 1: weaknesses)**
>
> Thank you for the thoughtful review and for highlighting the originality, efficiency, and broad empirical strength of SEKA. We truly appreciate the positive rating. We have added new analyses on data quantities influence, expanded the discussion on projection semantics, and clarified several methodological points. Below we respond to each of your comments in detail.
>
> **Weaknesses**
>
> > While the geometric interpretation is intuitive, the paper would be further strengthened by a linguistic or semantic characterization of the learned relevance subspace ...
>
> Thank you for raising this point. We agree that analysing nearest-neighbour semantics of the projection directions is interesting for steering-type of work. However, SEKA is not intended to manipulate or reshape semantic content. Its effect is deliberately confined to the attention-routing subspace, not the semantic subspace.
>
> This is supported by multiple existing works such as [1][2]. They show that:
> - Q/K vectors encode routing features that determine which tokens should attend to which.
> - Semantic meaning is primarily stored in V vectors and MLP activations.
>
> That's why existing works use activation steering to guide the generation toward a specific semantic or linguistic direction. But in our work this is fundamentally different as we aim to operate on the attention-routing subspace, not the semantic subspace. That’s exactly why we extract the projections from the same token spans. A geometric interpretation is therefore more appropriate for explaining our method than a linguistic analysis of projection directions. We sincerely thank you for raising this insightful comment as it reminds us we lack discussion on this part. **We have included the discussion and references in the revised version (at the end of Appendix C, lines 867-884**, will move to the main paper if we are allowed to have more space or an extra page).
>
> [1] A Mathematical Framework for Transformer Circuits. Anthropic.
>
> [2] In-context learning and induction heads. Anthropic.
>
> > ... Currently, experts are constructed based on different datasets, with no exploration of automatic clustering or dynamic expert generation.
>
> Thanks for raising this interesting point. Our empirical results in Table 2 already show that projections learned from synthetic contrastive prompts generalise broadly: vanilla SEKA, using only these universal projections, improves performance across multiple unseen tasks without any task-specific tuning. AdaSEKA further demonstrates that an automatic selection and weighting mechanism over several task-specific projections can sometimes bring additional gains even though the original purpose is to reduce the number of hyperparameters.
>
> We agree that automatic expert discovery through clustering or dynamic expert generation is an interesting next step. In practice, this will become more feasible as attention steering benchmarks grow to cover a wider range of task types. We also note an inherent trade-off: increasing the number of experts inevitably raises memory usage, so future work will need to balance automatic expert expansion with practical computational resource constraints.
>
> > Significant performance fluctuations in certain tasks ...
>
> Could you kindly point us where this behaviour was observed? If you are referring to the case where AdaSEKA outperforms SEKA on Gemma-3-4B but not on Gemma-3-12B. The motivation behind AdaSEKA is to reduce the number of hyperparameters by dynamically selecting task-specific projections. The results remain stable: both SEKA and AdaSEKA consistently improve over the original model across all tasks.
>
> If the fluctuation refers to the ablation “w/o learn”, this is unrelated to AdaSEKA, since that ablation reflects the absence of learned projections. In that case, we believe the concern arises from a misunderstanding.
>
> > It would be helpful to discuss the data efficiency of the offline spectral learning process, such as how projection stability changes with sample size.
>
> Thank you for this valuable suggestion. We completely agree. **We have therefore added a new analysis in Appendix K**, where we gradually increase the number of synthetic samples used for projection extraction and evaluate the resulting performance on all three benchmark tasks. The plots and discussion now appear in Appendix K (see **Figure 9**; unfortunately OpenReview does not allow us to attach the figure here). Below we summarise the key findings, which are discussed in detail in the appendix:
>
> - SEKA is generally data-efficient across models and tasks. Performance stabilises once roughly 50 synthetic samples are used.
> Increasing the number of samples consistently yield more stable behaviour.
> - Models within the same family exhibit similar qualitative trends, though model size still introduces noticeable differences.
>
> Thank you again for raising this point. It directly motivated us to expand the analysis and improve the robustness discussion in the revised draft.

---

> ### Author Response · Authors · 2025-11-20
> **Reply to Reviewer c6N4 (Part 2: Questions)**
>
> **Questions**
>
> > ... How do the authors justify that this specific construction of contrastive prompts can capture a sufficiently broad notion of relevance to generalize to unseen, and potentially more complex, tasks?
>
> Thank you for the insightful question. Our justification relies on two complementary pieces of evidence.
>
> First, the visualisations in Figure 6 (Appendix E) show that key embeddings for the same token spans exhibit a consistent shift from negative to positive variants across many heads. This confirms that the relevance direction exists and is consistent at certain heads, across different token. This is not task-specific as the ones used for AdaSEKA.
>
> Second, the quantitative results in Table 2 shows that vanilla SEKA, which only uses projections learned from this synthetic dataset and receives no task-specific tuning, generalises well to three diverse unseen tasks. This provides extra empirical support that the learned relevance subspace is sufficiently broad to transfer beyond the synthetic setting.
>
> Regarding more complex tasks, our paper focuses on benchmarks where the role of highlighted spans is well-defined, following the setup from previous work like PASTA [3] for fair comparisons. The current evidence already shows that our synthetic construction yields robust and transferable relevance signals across different tasks. Extending the approach to more structurally complex tasks (e.g., multi-hop reasoning, multi-turn dialogue, or long-context retrieval) is indeed an interesting direction, and might become more feasible as attention steering benchmarks grow to cover a wider range of task types.
>
> [3] Zhang, Qingru, et al. Tell Your Model Where to Attend: Post-hoc Attention Steering for LLMs. ICLR 2024.
>
> > Interpretability of Projection Directions.
>
> Please see our response to the related weakness above.
>
> > ... do the authors believe that combining SEKA with explicit positional bias calibration methods could yield further performance gains? Is this a worthwhile direction for future work?
>
> Thank you for raising this very insightful point. To draw concrete conclusions, further empirical work on combining SEKA with other calibration methods is definitely needed, but we can share our thoughts and clarify the relationship between SEKA and Found-in-the-Middle (FITM) from a linear-algebraic perspective.
>
> Both of them can be written in the same general form: $\text{Logits} = \frac{QK^\top}{\sqrt{d_k}} + B. $
>
> However, the nature of the bias term is different: FITM modifies a pure positional bias term $(B_{\text{FITM}})$, which depends only on token positions and is independent of the QK interaction. SEKA induces a bias
> $$B_{\text{SEKA}} = \frac{Q(g^+P^+K + g^-P^-K)^\top}{2\sqrt{d_k}},$$
>
> which is QK-dependent, and learned from contrastive relevance signals rather than positional structure. Because our projections are extracted from neutral/positive/negative variants of the same token spans, positional information is largely cancelled out when computing cross-covariances and performing SVD. What remains is the low-rank direction that consistently varies with relevance rather than with position. Thus we expect the correlation between SEKA’s relevance subspace and FITM’s positional-bias subspace to be very low, so that the two interventions should be largely complementary rather than redundant.
>
> That said, you might still need to pay attention when combining these two methods because applying both may over-amplify certain heads.
>
> In contrast, PASTA and SEKA are mathematically much closer. PASTA performs post-hoc row rescaling of the attention matrix, whereas SEKA achieves the similar end-effect by editing key vectors before attention computation. They both ultimately introduce a low-rank, relevance-dependent bias term onto the attention scores. So SEKA can be viewed as a more fine-grained and computationally efficient upgrade of PASTA, where the bias is not hand-crafted through row rescaling but learned explicitly through spectral decomposition of key-embedding shifts.
>
> > Could the source code be made publicly available?
>
> Yes, absolutely. As we mentioned in the Reproducibility Statement on page 10, all the code, preprocessed data, and pre-computed projections needed to reproduce our results will be made publicly available if the paper is accepted. We have also included information about the dataset license in that section to ensure everything is appropriate.
>
> ---
>
> We hope the clarifications and additional analyses address your concerns and further strengthen your overall assessment of the work. If any aspect would benefit from additional detail, we are happy to provide further clarification. Thanks again for your efforts in reviewing our paper!

---

### Comment · Area_Chair_28L5 · 2025-11-27
**Reviewer Reminder: Author Rebuttals Available**

Dear Reviewers,

The authors have posted their rebuttals to your reviews.

Please read the authors' responses, assess whether your concerns have been addressed, and update your ratings accordingly.

Your prompt attention to the rebuttals is appreciated.

Best,
AC

---

### Author Response · Authors · 2025-12-01
**Summary of Rebuttal Updates and Reviewer Consensus (Submission 7673)**

Dear Area Chair,

Given the recent disruptions and the resulting AC reassignment, we are writing to provide a concise summary of our rebuttal updates to assist in your assessment of our paper.

**1. General Consensus**

All four reviewers have assessed the paper positively (**Scores: 8, 6, 6, 6**), consistently highlighting the **originality** of our spectral editing approach and its **practical significance** (specifically, its full compatibility with Flash Attention and negligible latency compared to existing methods like PASTA).

**2. Key Rebuttal Actions & New Experiments**

We have addressed the primary questions raised by reviewers regarding robustness and implementation details. The revised paper includes extra analyses and clarifications:

*   **Data Efficiency Analysis (Added to Appendix K):** In response to Reviewers c6N4 and XAQm, we conducted new experiments varying the sample size of synthetic data. Results demonstrate that SEKA is highly data-efficient, stabilizing with as few as ~50 samples.
*   **Hyperparameter Sensitivity (Added to Appendix G):** Addressing concerns from Reviewers Z3wt and JuJy, we added a comprehensive sensitivity analysis of hyper-parameters. The results confirm that our method is  fairly robust. Most parameters have wide stable regions and only a couple require light tuning, which is also acknowledged by Reviewer JuJy, saying that **"Most of my concerns have been resolved."**
*   **Implementation Clarifications (Appendix A, B, & F.3):** We expanded the details on our lightweight synthetic data generation (which is fully automated) and clarified evaluation metrics regarding "empty responses" to resolve confusion raised by Reviewer XAQm.

**3. Current Status**

We believe these new experiments and revisions fully address the questions raised by the reviewers who have not yet been able to reply (3 out of 4 haven't responded). We hope this summary helps with your decision-making process.

We sincerely thank the reviewers for their constructive feedback. We are also deeply grateful to the Area Chair for overseeing this process. We know the recent circumstances have imposed a significant additional workload, and we truly appreciate your time and effort in reviewing our submission.

Best regards,

The Authors

---

### Meta-Review · Area_Chair_mfC1 · 2026-01-08

**Summary:**

The reviewers were generally positive about the paper, highlighting its originality, efficiency, and compatibility with modern attention mechanisms like Flash Attention. However, they raised several specific concerns regarding the methodology and evaluation, as the following.

Multiple reviewers (c6N4, Z3wt, JuJy) questioned the complexity and overhead of the offline spectral learning process. They expressed concern that constructing synthetic contrastive triplets might be tedious or that the method's effectiveness might hinge heavily on the quality and quantity of this data. There were questions about how projection stability changes with sample size.

Reviewers (JuJy, Z3wt) noted a lack of sensitivity analysis for the various hyperparameters introduced. The reviewer was concerned about the robustness of the method across different configurations.

Reviewer c6N4 asked for a semantic characterization of the learned subspaces beyond the geometric interpretation. Reviewer XAQm o the other hand, questioned the generalizability of projections learned on specific tasks to unseen instruction types.

**Reviewer Concerns:**

The authors provided a comprehensive rebuttal that addressed the majority of the reviewers' technical concerns through new analyses and clarifications.

Re: Data Efficiency: The authors added a new analysis investigating the relationship between the number of synthetic samples and performance. They demonstrated that the method is data-efficient. This directly addressed the concerns of c6N4 and XAQm.

Re: Hyperparameter Sensitivity: The authors included a new sensitivity analysis demonstrating that the method is fairly robust across a wide range of parameters, with consistent trends within model families. This addressed the lack of ablation studies cited by JuJy and Z3wt.

Re: Data Construction Pipeline: The authors clarified that the synthetic data generation is lightweight and fully automated via LLM prompting (Appendix A)).

Finally, the authors clarified that the vanilla SEKA method uses task-agnostic projections learned from synthetic data and generalizes well to unseen tasks without tuning, and also clarified the tunable nature of the positive/negative gains and explained the metric issues raised by Reviewer XAQm.

**Reviewer Scores:**

There wasn't much discussion in the system.

Reviewer c6N4 gave a 6. This reviewer appreciated the high originality and efficiency. The authors addressed their specific request for data efficiency analysis (Appendix K) and provided a strong justification regarding the semantic interpretation of the subspace.

Reviewer XAQm gave 8 that was already strong accept. The rebuttal clarified their questions regarding generalizability and data quantity. Their high score is likely solidified by the additional data efficiency analysis.

Reviewer Z3wt gave 6. The authors addressed this reviewer's main weakness regarding hyperparameter sensitivity by adding a new analysis in Appendix G. They also clarified the "steering all passages" behavior. Given that the authors provided the requested sensitivity data, this reviewer would likely maintain their positive assessment.

Reviewer JuJy gave 6: This reviewer explicitly commented post-rebuttal: "Most of my concerns have been resolved, so I have decided to keep my original score."

---

### Decision · Program_Chairs · 2026-01-26

Accept (Poster)